# PROVABLE WEAK-TO-STRONG GENERALIZATION VIA BENIGN OVERFITTING

**David X. Wu**
Department of EECS
UC Berkeley
Berkeley, CA 94720
david_wu@berkeley.edu

**Anant Sahai**
Department of EECS
UC Berkeley
Berkeley, CA 94720
sahai@eecs.berkeley.edu

## ABSTRACT

The classic teacher-student model in machine learning posits that a strong teacher supervises a weak student to improve the student's capabilities. We instead consider the inverted situation, where a weak teacher supervises a strong student with imperfect pseudolabels. This paradigm was recently brought forth by Burns et al. (2023) and termed *weak-to-strong generalization*. We theoretically investigate weak-to-strong generalization for binary and multilabel classification in a stylized overparameterized spiked covariance model with Gaussian covariates where the weak teacher's pseudolabels are asymptotically like random guessing. Under these assumptions, we provably identify two asymptotic phases of the strong student's generalization after weak supervision: (1) successful generalization and (2) random guessing. Our techniques should eventually extend to weak-to-strong multiclass classification. Towards doing so, we prove a tight lower tail inequality for the maximum of correlated Gaussians, which may be of independent interest. Understanding the multilabel setting reinforces the value of using logits for weak supervision when they are available.

## 1 INTRODUCTION

Motivated by the problem of aligning increasingly capable models, Burns et al. (2023) introduced the framework of *weak-to-strong generalization*[1], which draws an analogy between humans supervising superhuman AIs and weaker teacher models supervising stronger student models. This inverts the classic teacher-student model framework, which typically assumes that the teacher is stronger than the student. Burns et al. (2023) found that using GPT-2 to finetune GPT-4 can recover most of the performance of standard supervised finetuning with human-annotated data across standard NLP benchmarks, but struggles on more difficult tasks such as chess puzzles or reward modeling. One failure mode observed by Burns et al. (2023) is that the strong student sometimes learns to mimic the weak teacher, i.e. the strong model overfits to the limitations of the weaker one. Since Burns et al. (2023), this phenomenon has been studied in a variety of other empirical settings (see e.g. Ji et al. (2024); Guo et al. (2024); Liu & Alahi (2024); Yang et al. (2024); Tao & Li (2024)).

These empirical observations naturally lead to the question that we aim to answer in this paper: Can we identify a simple, concrete theoretical setting where we can provably exhibit different phases of weak-to-strong generalization? Below, we survey closely related areas of research.

**Pseudolabeling and synthetic data.** In semi-supervised learning, *pseudolabeling* refers to the method of using one model's outputs to generate pseudolabels for unlabeled data (Lee et al., 2013; Arazo et al., 2020; Rizve et al., 2021; Zhang et al., 2021; Cascante-Bonilla et al., 2021; He et al., 2024). These pseudolabels are then used to supervise the target model. We consider using the weak model to generate pseudolabels for the strong model in our concrete setting.

---

[1]This situation is related, but slightly different, from the earlier identified problem of easy-to-hard generalization (see e.g. Schwarzschild et al. (2021); Hase et al. (2024)) wherein a model is trained on "easy" cases but has to generalize to "hard" cases. The most important distinction is that in weak-to-strong generalization, the weak teacher actually gets things wrong.

Synthetic data generation is another widely popular paradigm for scaling up models in the absence of human-annotated data; see, e.g. (Nikolenko, 2021; Liu et al., 2024; Chen et al., 2021; Figueira & Vaz, 2022) and references therein. In this approach, one uses generative models to generate synthetic data, which can then be labeled and used to supervise other models. The use of synthetic data to train models resembles the weak-to-strong setup that we study, although we assume the unlabeled datapoints are generated from the ground-truth distribution. Nevertheless, we believe that extending our techniques may yield interesting insights on the success and failure modes of synthetic data.

**Supervised finetuning and scaling laws.** In contemporary deep learning, the dominant paradigm is to pretrain large (e.g. billions of parameters) models on copious (e.g. trillions of tokens) unlabeled data in a self-supervised fashion and then finetune the model on relatively tiny (e.g. less than 100K) finetuning datasets. The prevailing wisdom suggests that (1) pretraining helps the model learn useful features that can be repurposed for specific tasks and (2) the optimal size of models should scale with the amount of data (see, e.g. (Kaplan et al., 2020; Wei et al., 2021; Wang et al., 2022b; Liu et al., 2022; Hoffmann et al., 2022)). Much effort has gone into improving the effectiveness and efficiency of finetuning (Houlsby et al., 2019; Hu et al., 2021; Lester et al., 2021; Dettmers et al., 2024).

According to NTK theory, if the model weights do not move far from initialization during finetuning, we can approximate finetuning as training a generalized linear model using (tangent) features. These features arise from the gradients of the network's outputs with respect to the parameters at finetuning initialization. This motivates our studying the behavior of interpolating linear models in the context of supervised finetuning. Our concrete theoretical setting sits within the Gaussian-features linear-model style of Wei et al. (2022); Belkin et al. (2020); Mei & Montanari (2022); Bartlett et al. (2020); Muthukumar et al. (2020; 2021); Chatterji & Long (2021); Wang & Thrampoulidis (2021); Subramanian et al. (2022); Wang et al. (2022a); Wang et al. (2021); Cornacchia et al. (2023).

**Knowledge distillation.** The topic of weak-to-strong generalization is related to the extensive literature on knowledge distillation, which originated in the desire to compress large powerful models (teachers) into smaller ones (students) (Hinton et al., 2015; Buciluǎ et al., 2006). See Gou et al. (2021) for a general survey. Theoretical perspectives for the essentially underparameterized case were developed in Phuong & Lampert (2019); Ji & Zhu (2020), and the theoretically engaged literature in this area has continued to expand; see Yuan et al. (2024); Ojha et al. (2023); Safaryan et al. (2023); Alballa & Canini (2024); Hong et al. (2024); Sarnthein et al. (2023); Zhao & Zhu (2023); Das & Sanghavi (2023); Xu et al. (2024); Nagarajan et al. (2023); Borup & Andersen (2023); Harutyunyan (2023); Stanton et al. (2021); Mobahi et al. (2020) for a few representative more recent examples.

The question of how knowledge distillation can improve the student's generalization to surpass the teacher (especially when they have the same architecture) has led to the identification of several different underlying mechanisms (Yuan et al., 2024; Safaryan et al., 2023; Sarnthein et al., 2023; Das & Sanghavi, 2023; Nagarajan et al., 2023; Mobahi et al., 2020). Of these, the closest in spirit to our approach is the regularization viewpoint of Mobahi et al. (2020), which studies a kernel-regression model. They call out the crucial role of the spectrum of the Gram matrix — more specifically, self-distillation accentuates the importance of the larger eigenvalues, which has a regularizing effect by making the corresponding basis functions more prominent in the learned pattern.

**Concurrent theoretical work.** Zhang et al. (2024) engages with generative models and observes that temperature can play an important role in allowing a trained model to surpass its training sources. Somerstep et al. (2024) takes a transfer-learning perspective and asserts that naive fine-tuning on weak pseudolabels tends not to work; our results show conditions where this does in fact succeed. Charikar et al. (2024) takes a representation-centric perspective in a regression setting and zooms in on the question of how much better the representation is for the stronger model. Lang et al. (2024) studies the classification setting and takes a neighborhood perspective that posits that the stronger model's neighborhood structure allows it to average over the weak labels to get generalization. At a high level, our work along with Mobahi et al. (2020); Charikar et al. (2024); Lang et al. (2024) all circle around the idea that weak-to-strong generalization works when the cascading learning process purifies representations in the true direction and contracts in false directions.

## 1.1 CONTRIBUTIONS

In this paper, we explore weak-to-strong generalization in a stylized theoretical model that captures the dynamics of finetuning with weak supervision studied in Burns et al. (2023). Under a simple overparameterized spiked covariance model for the pretraining features, we prove that finetuning an overparameterized linear classifier using minimum $\ell_2$ norm interpolation on top of these features provably exhibits *two distinct phases* of weak-to-strong generalization. In particular, under certain scalings, as we increase the number of weakly labeled finetuning examples, the strong learner's asymptotic accuracy transitions from (1) random guessing to (2) perfect generalization; see Section 3 for a precise statement.

To be specific, we study the generalization of interpolating linear models with and without weak supervision. Although our results can be generalized to any weak teacher which produces logits using a linear head, we assume for the sake of concreteness that the weak teacher is also an interpolating linear model which can be fully expressed by the strong student. We discuss how our theoretical results connect to realistic supervised finetuning scenarios in Section 4.

Our results strongly hinge on the tight analysis for benign overfitting in multiclass classification in Wu & Sahai (2024), although the study of benign overfitting for regression and binary classification was already carried out by Bartlett et al. (2020); Muthukumar et al. (2021).

## 2 PRELIMINARIES AND SETUP

Below, we set up the weak-to-strong learning task in Section 2.1, along with the data assumptions in Section 2.2, and finally specify the concrete end-to-end learning algorithm we study in Section 2.3.

## 2.1 WEAK-TO-STRONG SETUP

To study weak-to-strong generalization, we will consider a simple setup which encapsulates the dynamics of standard supervised finetuning as well as training linear probes on intermediate activations. We will now give a high level description of the weak-to-strong setting, and in subsequent sections formally define the specific assumptions we make to theoretically study weak supervision.

Suppose we have $n$ labeled datapoints and $m = n^u$ unlabeled datapoints, where $u > 1$. This matches the modern ML paradigm where labeled data is scarce and unlabeled data is abundant.

We assume that we have access to two sets of features extracted from the datapoints: the weak and strong features. Using these features, we will create a weak-to-strong setup with a weak model and a strong model, where the weak model is used to generate hard pseudolabels to train the strong model (see Procedure 1 for a formal definition). The learning task at hand is binary classification, where each model uses its respective features obtained from the datapoints — see Section 4 for how our techniques apply to the multiclass setting.

Clearly, to get nontrivial learning guarantees, there needs to be some relationship between the weak and strong features. To specify this, we will assume that the weak and strong features come from an appropriate *weak-to-strong ensemble* of features, which we formally define in Section 2.2. One way to interpret the weak-to-strong ensemble is that the true hidden direction is highlighted more in the strong features than the weak features; see Figure 1.

In supervised finetuning and linear probing, the strong and weak features come from pretraining via the neural tangent features and intermediate activations, respectively. For example, in the GPT2 to GPT4 weak-to-strong setup of Burns et al. (2023), the weak features come from GPT2 pretraining, whereas the strong features come from GPT4 pretraining.

Broadly speaking, we study the case where the true labels are generated by a distinguished (but unknown) low-rank subspace hidden in very high dimensional space.

To be concrete, we will study two different classifiers in this setup:

(1) $f_{\text{weak}}$: train/finetune on $n$ datapoints using weak features in $\mathbb{R}^{d_{\text{weak}}}$ and ground-truth labels.

(2) $f_{\text{w2s}}$: train/finetune on $m \gg n$ datapoints using strong features in $\mathbb{R}^d$ and hard pseudolabels generated from $f_{\text{weak}}$.

We study a scheme where $\boldsymbol{f}_{\text{weak}}$ and $\boldsymbol{f}_{\text{w2s}}$ are linear models trained by performing minimum $\ell_2$ interpolation (MNI) on their respective training sets; see Section 2.3 for a formal definition of MNI.

**Remark 2.1.** *Instead of training with hard (categorical) pseudolabels, one could use the real-valued scores from $\boldsymbol{f}_{\text{weak}}$. This would only affect constants which are not crucial to any of our results.*

We will measure generalization via the test accuracy of these classifiers. In particular, let $\ell(\cdot)$ be the 0-1 loss function, and let $\mathbf{E}[\ell(\cdot)]$ be the expected test error over a fresh test sample. We introduce the following desiderata to define weak-to-strong generalization for binary classification.

**Desiderata 1.** *The main desiderata are the following:*

(i) *The strong model asymptotically generalizes[2] when trained on the $m$ weakly labeled data-points: $\mathbf{E}[\ell(\boldsymbol{f}_{\text{w2s}})] = o_n(1)$.*

(ii) *The strong model can fully represent the weak model.*

(iii) *The weak model asymptotically does not generalize: $\mathbf{E}[\ell(\boldsymbol{f}_{\text{weak}})] = \frac{1}{2} - o_n(1)$.*

In Section 3, we show that the above desiderata are achievable in a simple toy model; see Theorem 3.3 for a formal statement. This paints a rather striking picture: there are situations where the weak labels asymptotically look like random guessing (Desideratum 1.iii), the strong model can perfectly imitate the weak labels (Desideratum 1.ii), yet the strong model still asymptotically generalizes by extracting enough signal out of the plentiful weak labels (Desideratum 1.i).[3]

We also include some bonus desiderata, which paint a comparison to natural alternative training methods. We can also provably achieve these bonus desiderata in certain regimes; see Remark 3.4.

**Desiderata 2.** *The extra desiderata are the following:*

(i) *PCA cannot recover the low-rank structure from $n + m$ observations of the strong features.*

(ii) *Let $\boldsymbol{f}_{\text{strong}}$ be the strong model when trained on $n$ datapoints using strong features in $\mathbb{R}^d$ and ground-truth labels. Then $\boldsymbol{f}_{\text{strong}}$ asymptotically fails: $\mathbf{E}[\ell(\boldsymbol{f}_{\text{strong}})] = \frac{1}{2} - o_n(1)$.*

## 2.2 DATA MODEL

Throughout, we will consider data with zero mean Gaussian covariates, which can be viewed as an idealized version of the pretraining features or representations. These covariates are generated from an ambient standard Gaussian vector $\boldsymbol{g} = (g_1, \ldots, g_D) \in \mathbb{R}^D$. We make Gaussianity assumptions for the sake of theoretical tractability; in Section 4 we discuss potential extensions to other settings.

**Covariates.** A learner observes iid features $\boldsymbol{x}_i \sim N(0, \Sigma)$, where we emphasize that the feature covariances $\Sigma \in \mathbb{R}^{d \times d}$ are unknown and different for each learner. The different $\Sigma$'s capture how weak or strong the features are for each model; we make this precise below. Note that $\boldsymbol{x}_i$ is a linear transformation of the underlying randomness $\boldsymbol{g}_i \sim N(0, I_D)$. We will often refer to the eigendecomposition $\Sigma = U\Lambda U^\top$, where $U \in \mathbb{R}^{d \times d}$ is orthogonal and $\Lambda \in \mathbb{R}^{d \times d}$ is diagonal.

**Labels.** For binary classification, we consider hard labels generated by the signs of Gaussians, so that $y = \text{sgn}(\langle \boldsymbol{g}, \boldsymbol{v}_* \rangle)$, where $\boldsymbol{v}_*$ is an unknown unit-norm direction. To analyze the generalization of various learners, we will allow ourselves to study labels generated by various directions $\boldsymbol{v}$, not just the true $\boldsymbol{v}_*$, all of which we assume are unknown.[4] For multiclass classification, we assume the labels are generated via $y = \arg\max_{j \in [k]} \langle \boldsymbol{g}, \boldsymbol{v}_*^{(j)} \rangle$, where $\boldsymbol{v}_*^{(1)}, \ldots, \boldsymbol{v}_*^{(k)}$ are all unknown.

Following Muthukumar et al. (2021); Subramanian et al. (2022); Wu & Sahai (2024), we will make the following assumption on the true label directions to simplify the analysis. However, to study weak-to-strong generalization, we will eventually have to analyze a weak label direction which is *not* 1-sparse. One of our main contributions is showing how to analyze this case.

---

[2]The natural extension to multiclass settings with $k$ different classes would require $\mathbf{E}[\ell(\boldsymbol{f}_{\text{weak}})] = 1 - \Theta(\frac{1}{k})$.

[3]In particular, Desideratum 1.ii allows the failure mode of the strong model imitating the weak model and also rules out more trivial sources of weak labels, such as independent label noise. If the strong model cannot represent the noise in the weak labels, then weak-to-strong generalization is intuitively much simpler.

[4]To study weak supervision, we take $\boldsymbol{v} = \boldsymbol{w}_{\text{weak}}$, the direction that the weak model learns.

**Assumption 1** (1-sparse assumption). *We say that the labels satisfy the 1-sparse assumption relative to covariance $\Sigma$ if the following holds. The label defining direction $\boldsymbol{v}_*$ (directions $\boldsymbol{v}_*^{(1)}, \ldots, \boldsymbol{v}_*^{(k)}$ for multiclass) is aligned with a top eigenvector (top-$k$ eigenvectors for multiclass) of $\Sigma$ such that, in an eigenbasis where $\boldsymbol{v}_* = \boldsymbol{e}_1$ and $\boldsymbol{v}_*^{(i)} = \boldsymbol{e}_i$, respectively, we have*

$$y = \operatorname{sgn}(x_1) \qquad\qquad\qquad \text{(Binary)}$$

$$y = \arg\max_{j \in [k]} x_j \qquad\qquad\qquad \text{(Multiclass)}$$

For example, if the 1-sparse assumption holds for the strong covariance, then in a strong eigenbasis where the strong features are independent, the labels are generated by axis-aligned directions.

**Bi-level ensemble and weak-to-strong ensemble.** To simplify the analysis, we follow Muthukumar et al. (2021); Subramanian et al. (2022); Wu & Sahai (2024) and assume the eigenvalues $\Lambda$ of the covariance are parameterized by the bi-level ensemble defined shortly. The bi-level ensemble is a simple overparameterized version of the well-known spiked covariance model for PCA.

**Definition 1** (Bi-level ensemble). *Let $\boldsymbol{x} \sim N(0, \Sigma)$, where the covariance $\Sigma = U\Lambda U^\top$. The bi-level ensemble parameterizes $\Lambda = \Lambda(p, q, r)$, where $p > 1$, $0 \leqslant r < 1$, and $0 < q < (p - r)$. The number of features ($d$), number of spiked directions ($s$), and degree of favoring ($a$) all scale with the number of training points ($n$) as follows:*

$$d = \lfloor n^p \rfloor, s = \lfloor n^r \rfloor, a = n^{-q}. \tag{1}$$

*Then $\Lambda = \operatorname{diag}(\lambda_i)_{i \in [d]}$, where*

$$\lambda_j = \begin{cases} \frac{ad}{s}, & 1 \leqslant j \leqslant s \\ \frac{(1-a)d}{d-s}, & \text{otherwise} \end{cases}. \tag{2}$$

*If the above holds, we refer to $\boldsymbol{x}$, $\Lambda$, or $\Sigma$ as being drawn from the bi-level ensemble. We use the shorthands $\lambda_F \triangleq \frac{ad}{s}$ and $\lambda_U \triangleq \frac{(1-a)d}{d-s}$ to denote the favored and unfavored eigenvalues, respectively.*

In particular, the parameterization controls the total number of features, $d = n^p \gg n$, as well as the dimension of the low-rank subspace $s = n^r \ll n$. For multiclass classification, we allow the number of classes to also scale with the number of datapoints $n$.

**Definition 2** (Scaling for multiclass). *For multiclass classification with $k$ classes, we have $k = c_k \lfloor n^t \rfloor$, where $0 \leqslant t < r$ and $c_k$ is a positive integer.*

We now pin down a concrete weak-to-strong ensemble, which boils down to specifying the joint distribution of the weak and strong features. We impose a subset relationship between the weak and strong features in the strong basis; see Figure 1 for a diagram.

**Assumption 2** (Weak-to-strong subset ensemble). *Let $\Lambda = \Lambda(p, q, r) \in \mathbb{R}^{d \times d}$ denote the strong eigenvalues and $\Lambda_{\text{weak}} = \Lambda(p_{\text{weak}}, q_{\text{weak}}, r_{\text{weak}}) \in \mathbb{R}^{d_{\text{weak}} \times d_{\text{weak}}}$ denote the weak eigenvalues, both drawn from the bi-level ensemble. Let $\lambda_{F,\text{weak}} \triangleq \frac{a_{\text{weak}} d_{\text{weak}}}{s_{\text{weak}}}$ and $\lambda_{U,\text{weak}} \triangleq \frac{(1-a_{\text{weak}}) d_{\text{weak}}}{d_{\text{weak}} - s_{\text{weak}}}$ denote the weak favored and unfavored eigenvalues, respectively. Let $U$ be any distinguished eigenbasis of $\Sigma$ where $\boldsymbol{v}_* = \boldsymbol{e}_1$ (Assumption 1). The weak and strong features in the basis $U$ are related as follows.*

(1) *$\boldsymbol{x}_{\text{strong}} \sim N(0, \Lambda)$, where $\Lambda = \lambda_F I_{[s]} + \lambda_U I_{[d] \setminus [s]}$.*

(2) *There exists subsets of coordinates $S \subseteq [s], T \subseteq [d] \setminus [s]$, with $1 \in S$ and $|S| = s_{\text{weak}}$, such that*

$$\boldsymbol{x}_{\text{weak}} = \left(\sqrt{\frac{\lambda_{F,\text{weak}}}{\lambda_F}} \Pi_S + \sqrt{\frac{\lambda_{U,\text{weak}}}{\lambda_U}} \Pi_T\right) \boldsymbol{x}_{\text{strong}} \stackrel{d}{=} N(0, \lambda_{F,\text{weak}} I_S + \lambda_{U,\text{weak}} I_T).$$

*Here, $\Pi_S$ denotes the projection onto the axis-aligned subspace indexed by $S$.*

*We will often abuse notation and restrict $\boldsymbol{x}_{\text{weak}} \in \mathbb{R}^d$ to the coordinates in $S \cup T$, viewing $\boldsymbol{x}_{\text{weak}} \in \mathbb{R}^{d_{\text{weak}}}$. Hence, the weak favored features are a subset of the strong favored features, the weak unfavored features are a subset of the strong unfavored features, with different bi-level scalings, and furthermore the 1-sparse assumption holds for both $\Sigma$ and $\Sigma_{\text{weak}}$.*

Clearly, under Assumption 2, desideratum 1.ii is satisfied, and the subset ensemble is essentially the simplest relationship between the features one could impose to achieve the desideratum. In Section 4, we discuss how we expect the results to change if we relax these data modeling assumptions.

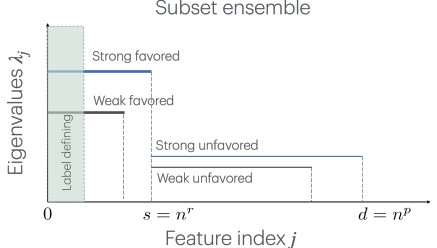

Figure 1: Visualization of subset ensemble (Assumption 2) relating weak and strong features. Notice the decreased favoring for the weak features, and how the weak features are a subset of the strong features in their respective category. Hence, a linear model on the strong features can simulate one on the weak features (Desideratum 1.ii). Moreover, the label defining directions, represented by the green shaded box, are in the span of both the weak and strong features.

## 2.3 LEARNING ALGORITHM

Our models are all trained using minimum $\ell_2$-norm interpolation (MNI), which corresponds to the asymptotic behavior of gradient descent with zero initialization (Ji & Telgarsky, 2021). Define the data matrix $\boldsymbol{X} \in \mathbb{R}^{n \times d}$ by $\boldsymbol{X}^\top \triangleq [\boldsymbol{x}_1 \quad \cdots \quad \boldsymbol{x}_n]$. Let $\boldsymbol{y}^{(i)} \in \mathbb{R}^n$ be the (centered) one-hot vector for class $i \in [k]$. In MNI, we learn a linear score function $\boldsymbol{f}^{(i)}$ via the optimization problem

$$\boldsymbol{f}^{(i)} = \arg\min_{\boldsymbol{f}} \|\boldsymbol{f}\|_2 \qquad \text{s.t. } \boldsymbol{X}\boldsymbol{f} = \boldsymbol{y}^{(i)}. \tag{MNI}$$

The coefficients of $\boldsymbol{f}^{(i)}$ can be easily computed to be $\boldsymbol{f}^{(i)} = \boldsymbol{X}^\top(\boldsymbol{X}\boldsymbol{X}^\top)^{-1}\boldsymbol{y}^{(i)} \triangleq \boldsymbol{X}^\top \boldsymbol{A}^{-1}\boldsymbol{y}^{(i)}$, where the matrix $\boldsymbol{A} \triangleq \boldsymbol{X}\boldsymbol{X}^\top \in \mathbb{R}^{n \times n}$ is the unnormalized Gram matrix and invertible almost surely.

At test time, given a fresh sample $\boldsymbol{x}_{\text{test}} \sim N(0, \Sigma)$, in the binary case we predict $\widehat{y} = \text{sgn}(\langle \boldsymbol{f}, \boldsymbol{x}_{\text{test}} \rangle)$, and for multiclass we predict the class with the highest score: $\widehat{y} = \arg\max_{i \in [k]} \langle \boldsymbol{f}^{(i)}, \boldsymbol{x}_{\text{test}} \rangle$. To be very explicit, let us formally define the end-to-end traning procedure for weak-to-strong binary classification (cf. the description before Remark 2.1), with an obvious extension to multiclass settings.

**Procedure 1** (Weak-to-strong training). *The weak learner observes an initial dataset of $n$ datapoints $(\widetilde{\boldsymbol{x}}_{i,\text{weak}}, y_i)_{i \in [n]}$, where $\widetilde{\boldsymbol{x}}_{i,\text{weak}}$ are the weak features for the ith datapoint and $y_i = \text{sgn}(\langle \boldsymbol{g}_i, \boldsymbol{v}_* \rangle)$ is the corresponding clean hard label. We train $\boldsymbol{f}_{\text{weak}} \in \mathbb{R}^{d_{\text{weak}}}$ using MNI on these $n$ clean datapoints.*

*Then, both learners observe $m$ extra unlabeled datapoints, where the weak model sees weak features $(\boldsymbol{x}_{j,\text{weak}})_{j \in [m]}$ and the strong model sees the corresponding strong features $(\boldsymbol{x}_{j,\text{strong}})_{j \in [m]}$. Generate $m$ hard pseudolabels via $\widehat{y}_{j,\text{weak}} = \text{sgn}(\langle \boldsymbol{f}_{\text{weak}}, \boldsymbol{x}_{j,\text{weak}} \rangle)$, and use MNI to train $\boldsymbol{f}_{\text{w2s}} \in \mathbb{R}^d$ on $(\boldsymbol{x}_{j,\text{strong}}, \widehat{y}_{j,\text{weak}})_{j \in [m]}$.*

## 3 MAIN RESULTS

In this section, we state our main results for binary classification (Theorem 3.3) and multilabel classification (Theorem 3.5). We focus on the regime $q + r > u$ and $q_{\text{weak}} + r_{\text{weak}} > 1$, since Muthukumar et al. (2021, Theorem 13) implies that these are the only nontrivial regimes for binary classification under the bi-level ensemble.

In this regime, we also tighten the previous error rates for binary and multiclass classification in the bi-level ensemble from (Wu & Sahai, 2024, Theorem 3.2); the proof can be found in Appendix E. To that end, we establish a new concentration inequality for the lower tail of the maximum of correlated Gaussians, which may be of independent interest; we defer its statement to the end of this section.

**Theorem 3.1** (Regimes with clean labels). *Suppose the strong features have bi-level covariance $\Sigma = \Sigma(p, q, r)$ (Definition 1), where $q + r > 1$, the true multiclass labels are 1-sparse (Assumption 1), and the number of classes $k = \lfloor n^t \rfloor$ (Definition 2). Then the test error for $\boldsymbol{f}_{\text{strong}}$ MNI-trained with $n$*

*clean multiclass labels satisfies*

$$\mathbf{E}[\ell(\boldsymbol{f}_{\text{strong}})] = \begin{cases} o_n(1), & \text{if } t < \min\{1-r, \tau_{\text{strong}}\} \\ 1 - \Theta(\frac{1}{k}), & \text{if } t > \min\{1-r, \tau_{\text{strong}}\} \end{cases}, \tag{3}$$

*where* $\tau_{\text{strong}} \triangleq p + 1 - 2(q+r)$. *Furthermore, for binary classification (i.e.* $k = 2$), *the explicit error rates are given by*

$$\mathbf{E}[\ell(\boldsymbol{f}_{\text{strong}})] = \frac{1}{2} - \frac{1}{\pi} \arctan(\Theta(n^{\tau_{\text{strong}}})). \tag{4}$$

By generalizing other results from Wu & Sahai (2024) to handle imperfect labels, we formally establish weak-to-strong generalization for binary classification in the subset ensemble. We give a brief technical overview of the proof in Section 3.1 and prove it formally in Appendix C.

**Theorem 3.2** (Weak-to-strong generalization for subset ensemble). *Consider the setup in Procedure 1 where the weak model* $\boldsymbol{f}_{\text{weak}}$ *is MNI-trained on* $n$ *correctly labeled examples and the strong model* $\boldsymbol{f}_{\text{w2s}}$ *is MNI-trained on* $m = n^u$ *weakly pseudolabeled examples, where* $q + r > u$ *and* $q_{\text{weak}} + r_{\text{weak}} > 1$. *In addition, we make the following data assumptions:*

(1) *The true binary labels are 1-sparse (Assumption 1) for the strong covariance.*

(2) *The weak and strong features follow the subset ensemble (Assumption 2) with bi-level eigenvalues* $\Lambda(p, q, r)$ *and* $\Lambda(p_{\text{weak}}, q_{\text{weak}}, r_{\text{weak}})$, *respectively, scaled relative to* $n$.

(3) *There are not too many weakly labeled examples:* $u < \frac{p+1+q+r-(q_{\text{weak}}+r_{\text{weak}})}{2}$.

*Recall* $\tau_{\text{strong}} \triangleq p + 1 - 2(q+r)$. *Then, the resulting test error for* $\boldsymbol{f}_{\text{w2s}}$ *satisfies*

$$\mathbf{E}[\ell(\boldsymbol{f}_{\text{w2s}})] = \begin{cases} o_n(1), & \text{if } u > q_{\text{weak}} + r_{\text{weak}} - \min\{1-r, \tau_{\text{strong}}\} \\ \frac{1}{2} - o_n(1), & \text{if } u < q_{\text{weak}} + r_{\text{weak}} - \min\{1-r, \tau_{\text{strong}}\}. \end{cases} \tag{5}$$

In other words, under the stated parameter regimes, weak-to-strong generalization occurs when the strong model is trained on sufficiently many weak binary labels.[5] Figure 2 demonstrates that the conditions for weak-to-strong generalization are not vacuous. We make the third enumerated assumption for technical reasons, but we believe this condition is essentially tight.

By combining Theorems 3.1 and 3.2, we obtain the exact conditions where the main desiderata from Section 2 are satisfied in the bi-level ensemble. The regimes where our theorem applies are depicted in Figures 2a and 2b, and we validated our theory with numerical simulations of MNI with $n = 50$ in Figures 2c and 2d; see Appendix F for more details on the experiments.

**Theorem 3.3** (Main result). *Under the same setup as Theorem 3.2, all the core weak-to-strong desiderata Desiderata 1.i to 1.iii are satisfied under the following conditions:*

$$u > q_{\text{weak}} + r_{\text{weak}} - \min\{1-r, \tau_{\text{strong}}\} \qquad \text{(Weak-to-strong generalization)}$$
$$0 > \tau_{\text{weak}}, \qquad \text{(Weak model does not generalize)}$$

*where* $\tau_{\text{weak}} \triangleq p_{\text{weak}} + 1 - 2(q_{\text{weak}} + r_{\text{weak}})$ *and* $\tau_{\text{strong}} \triangleq p + 1 - 2(q+r)$. *In particular, the conditions on the number of weak examples* $m = n^u$ *are non-vacuous as long as*

$$\tau_{\text{w2s}} \triangleq p + 1 - (q + r + q_{\text{weak}} + r_{\text{weak}}) > 0.$$

Note that under Assumption 2, Desideratum 1.ii is immediately satisfied since the weak features are spanned by the strong features. Also, as a sanity check, we confirm that whenever weak-to-strong generalization occurs with $m$ weak labels, the strong model would generalize with $m$ clean labels. By Theorem 3.1, if the strong model was instead trained on $m = n^u$ clean examples, it would generalize whenever $u > 2(q+r) - p$. By expanding the definition of $\tau_{\text{strong}}$ and using $q_{\text{weak}} + r_{\text{weak}} > 1$,

$$u > q_{\text{weak}} + r_{\text{weak}} - \tau_{\text{strong}} = 2(q+r) - p + (q_{\text{weak}} + r_{\text{weak}} - 1) > 2(q+r) - p,$$

which recovers the condition with $m$ clean labels stated above.

---

[5]We expect weak imitation to occur as soon as $m \gg d$ if the models are trained using gradient descent, i.e. the classic underparameterized regime with more weakly labeled datapoints than features.

**Remark 3.4** (Bonus desiderata). *If $q + r > u$, as in the setup of Theorem 3.3, PCA fails to recover the spiked subspace (Shen et al., 2013; Fan & Wang, 2015), so the bonus Desideratum 2.i automatically holds. Additionally, by using the condition for where the strong model fails with $n$ clean examples from Theorem 3.1 ($\tau_{\text{strong}} < 0$), one can construct parameter regimes where the bonus Desideratum 2.ii also holds. The theorem also implies the strong model cannot bootstrap its own performance. Indeed, some basic algebra shows $\tau_{\text{w2s}} = \tau_{\text{strong}} = \tau_{\text{weak}}$ if $(p, q, r) = (p_{\text{weak}}, q_{\text{weak}}, r_{\text{weak}})$.*

We can extend the above result to the *multilabel* setting, which is a variant of multiclass classification where a datapoint can have multiple positive labels. The basic idea is that multilabel training can be approximated as several independent binary classification problems, and then we can apply the binary analysis. Since this requires additional setup, we defer the formal details to Appendix D.

**Theorem 3.5** (Informal, see Theorem D.3). *Under the analogous assumptions as Theorem 3.3 for multilabel classification, there is weak-to-strong generalization for $f_{\text{w2s}}$ trained on $m$ weakly hard multilabeled examples from $f_{\text{weak}}$ in the exact same regimes.*

The above result also suggests an interesting result for true multiclass problems with one-hot labels. In particular, one can use the multilabel scheme to generate weak multilabels for a *multiclass* weak-to-strong classifier $f_{\text{w2s}}$. One can argue that whenever multilabel weak-to-strong generalization occurs, so too does multiclass weak-to-strong generalization (see Appendix D.1 for more justification). By comparing to the regimes for clean multiclass labels due to Theorem 3.1, there exists regimes where the strong model would fail if trained using the same number of *clean multiclass* labels (i.e. only one positive label per example). At a high level, this can happen because there are too many classes, which sparsifies the label vectors. In contrast, the weak multilabels do not suffer from the sparsity issue, even though the underlying signal is noisier. This appears to be related to the strategy of using soft labels or logits in the pseudolabeling and knowledge distillation literature.

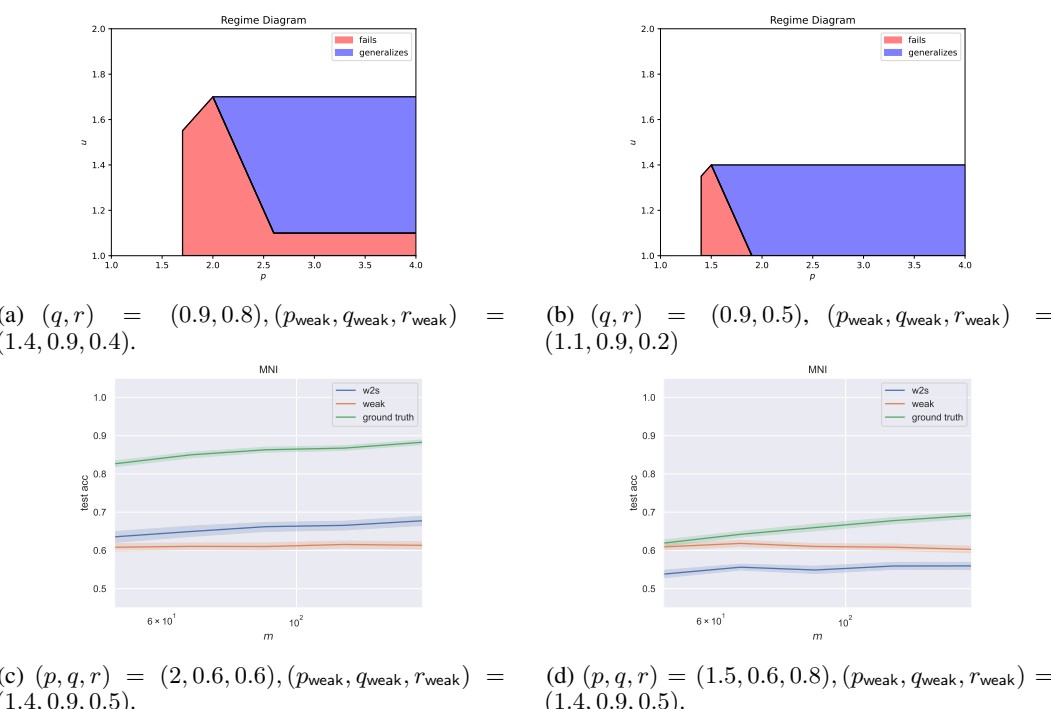

(a) $(q, r) = (0.9, 0.8), (p_{\text{weak}}, q_{\text{weak}}, r_{\text{weak}}) = (1.4, 0.9, 0.4)$.

(b) $(q, r) = (0.9, 0.5), (p_{\text{weak}}, q_{\text{weak}}, r_{\text{weak}}) = (1.1, 0.9, 0.2)$

(c) $(p, q, r) = (2, 0.6, 0.6), (p_{\text{weak}}, q_{\text{weak}}, r_{\text{weak}}) = (1.4, 0.9, 0.5)$.

(d) $(p, q, r) = (1.5, 0.6, 0.8), (p_{\text{weak}}, q_{\text{weak}}, r_{\text{weak}}) = (1.4, 0.9, 0.5)$.

Figure 2: **(Top)**: Regime plots for weak-to-strong generalization based on Theorem 3.3. The blue region is the successful w2s regime, and the red region is where w2s training fails. The white region corresponds to regimes where either the hypotheses of the theorem fail to hold, or invalid settings of parameters for the bi-level ensemble. **(Bottom)**: Comparison of simulations of MNI test accuracies for two different regimes where $n = 50$. Observe how the weak accuracy is close to random guessing and how the weak-to-strong accuracy increases as $m$ increases. As corroborated by the plots, Theorem 3.3 predicts w2s success in Figure 2c and failure in Figure 2d.

Finally, we state the aforementioned concentration inequality for the maximum of correlated Gaussians, which may be of independent interest. Our result complements the sharp bound of Lopes & Yao (2022) for more moderate deviations; the proof can be found in Appendix E.

**Theorem 3.6** (Lower tail for correlated Gaussians). *Let $\rho_0 \in (0,1)$ be a parameter bounded away from 0 and 1, and let $(g_i)_{i \in [N]}$ be jointly Gaussian with zero mean and unit variance. Suppose $\mathbf{E}[g_i g_j] \leqslant \rho_0$ for all distinct $i, j \in [N]$. For any $0 \leqslant t_N = \delta_0 \sqrt{2(1-\rho_0) \log N}$ where $\delta_0 \in [0,1)$ is bounded away from 1, there is a constant $C > 0$ depending only on $\rho_0$ such that*

$$\mathbf{Pr}\left[\max_{i \in [N]} g_i \leqslant t_N\right] \leqslant C \cdot N^{(1-\delta_0)^2(1-\frac{1}{\rho_0})} (\log N)^{\frac{1-\rho_0(2-\delta_0)-\delta_0}{2\rho_0}}.$$

*In particular, one can take $C = \sqrt{\frac{\rho_0}{1-\rho_0}}$. If we further have $\mathbf{E}[g_i g_j] = \rho_0$ for all distinct $i, j$ and $t_N = O\left(\frac{\log \log N}{\sqrt{\log N}}\right)$, then $\mathbf{Pr}\left[\max_{i \in [N]} g_i \leqslant t_N\right] = \Theta\left(N^{1-\frac{1}{\rho_0}} (\log N)^{\frac{1}{2\rho_0}-1}\right).$*

### 3.1 TECHNICAL OVERVIEW

In this section, we illustrate the main ideas of the proof of Theorem 3.2. Recall that the true label is $\mathrm{sgn}(x_*)$, where $x_* = \langle g_{\text{test}}, v_* \rangle$. Thus, studying weak-to-strong generalization amounts to analyzing

$$\mathbf{Pr}[\mathrm{sgn}(\langle f_{\text{w2s}}, x_{\text{test}} \rangle) = \mathrm{sgn}(x_*)], \tag{6}$$

where the fresh test point $x_{\text{test}} \sim N(0, \Sigma)$. Observe that conditioned on the training data $X$, the true Gaussian $x_*$ and $\langle f_{\text{w2s}}, x_{\text{test}} \rangle$ are jointly Gaussian random variables, so there exist $\lambda \in \mathbb{R}$ and $g \sim N(0,1)$ independent of $x_*$ such that

$$\langle f_{\text{w2s}}, x_{\text{test}} \rangle = \lambda x_* + g. \tag{7}$$

The exact value of $\lambda$ depends on exactly how much of $x_*$'s signal survives through the weak-to-strong training process. Intuitively, the larger $\lambda$ is, the higher the accuracy. Indeed, it is well-known from the classical study of noise stability that the probability in Equation (6) is precisely $\frac{1}{2} + \frac{1}{\pi} \arctan(\lambda)$.

To compute $\lambda$, we use Gram-Schmidt to decompose $f_{\text{w2s}}$ to obtain the signal and noise with respect to $x_*$. We term these quantities the survival and contamination of $x_*$, respectively. Without loss of generality, we can rotate to the basis $U$ where the strong features are drawn iid from $N(0, \Lambda)$, where $\Lambda = \mathrm{diag}(\lambda_1, \ldots, \lambda_d)$. After applying the basis change, one easily computes $\langle f_{\text{w2s}}, x_{\text{test}} \rangle \overset{d}{=} N(0, \|\Lambda f_{\text{w2s}}\|_2^2)$. The survival and contamination are then defined as follows.

**Definition 3** (Survival and contamination). *Let $v \in \mathbb{R}^D$ be a unit vector and let $f \in \mathbb{R}^d$ be a linear classifier for features drawn from $N(0, \sum_{i \in [d]} \lambda_i v_i v_i^\top)$, where $v_1, \ldots, v_d \in \mathbb{R}^D$ are orthonormal. The survival and contamination of $v$ in $f$ is defined as*

$$\mathsf{SU}(v) \triangleq \sum_{i \in [d]} \sqrt{\lambda_i} f[i] \langle v_i, v \rangle, \qquad \mathsf{CN}(v) \triangleq \sqrt{\|\Lambda f\|_2^2 - \mathsf{SU}(v)^2}.$$

*If $f$ is trained by MNI on $y = \mathrm{sgn}(\langle g, w \rangle)$, we write $\mathsf{SU}_n(v|w)$ for the survival of $v$ given $n$ labels generated from $w$, and similarly for $\mathsf{CN}_n(v|w)$. If $n$ or $w$ is clear from context, we omit them.*

To interpret these notions, consider the simple case where $v = e_1$ and $v_i = e_i$, which is the 1-sparse setting studied by previous papers. There, the survival is simply just $\sqrt{\lambda_1} f[1]$ and the contamination is $\sqrt{\sum_{i>1} \lambda_i f[i]^2}$; this was used to analyze binary classification in (Muthukumar et al., 2021).

In the weak-to-strong setting, we have $\langle f_{\text{w2s}}, x_{\text{test}} \rangle = \mathsf{SU}_m(v_*|v_{\text{weak}}) x_* + \mathsf{CN}_m(v_*|v_{\text{weak}}) g$, where $v_{\text{weak}} = f_{\text{weak}}/\|f_{\text{weak}}\|$ is the unit-norm direction learned by the weak model, and $g$ is a standard Gaussian independent of $x_*$. Plugging in $x_* = \langle g, v_* \rangle$ into (7) yields

$$\mathbf{Pr}[\mathrm{sgn}(\mathsf{SU}_m(v_*|v_{\text{weak}}) x_* + \mathsf{CN}_m(v_*|v_{\text{weak}}) g) = \mathrm{sgn}(x_*)], \tag{8}$$

from which we can read off $\lambda = \frac{\mathsf{SU}_m(v_*|v_{\text{weak}})}{\mathsf{CN}_m(v_*|v_{\text{weak}})}$. Thus, the explicit formula for the probability yields that $f_{\text{w2s}}$ approaches perfect accuracy if and only if $\frac{\mathsf{SU}_m(v_*|v_{\text{weak}})}{\mathsf{CN}_m(v_*|v_{\text{weak}})} = \omega(1)$, whereas its accuracy approaches random guessing if and only if $\frac{\mathsf{SU}_m(v_*|v_{\text{weak}})}{\mathsf{CN}_m(v_*|v_{\text{weak}})} = o(1)$.

We now turn to the strategy for controlling the survival and contamination. Using the explicit MNI formula for $f_{\text{weak}}$ and $f_{\text{w2s}}$, this analysis reduces to controlling certain random bilinear forms. Applying standard eigenvalue concentration results and a version of the Hanson-Wright concentration inequality for bilinear forms developed by Wu & Sahai (2024), we can precisely control the coordinates of $f_{\text{weak}}$, and hence $v_{\text{weak}}$. Once we have control over $v_{\text{weak}}$, we can then study $f_{\text{w2s}}$.

One technical difficulty is that the weak labels are no longer 1-sparse, even if the true labels are. This marks a departure from previous analyses of benign overfitting in classification, which all assumed 1-sparse labels (Muthukumar et al., 2021; Wang et al., 2021; Wu & Sahai, 2024). To overcome this obstacle, we develop tools to analyze MNI beyond the 1-sparse regime, namely by using the Woodbury inversion formula and carefully bounding error terms using Hanson-Wright. Under the subset ensemble, the explicit parameterizations for the survival and contamination then yield the stated conditions for weak-to-strong generalization.

## 4 DISCUSSION

We now discuss further extensions which suggest interesting new weak-to-strong phenomena.

**Modeling assumptions.** In this paper, we assumed the features were Gaussian for the sake of theoretical tractability. One could relax this to vector-subgaussian, as this notion is preserved under rotation. On the other hand, the bi-level and subset ensembles were chosen as a minimal set of tractable theoretical conditions to study weak-to-strong generalization. In principle, our techniques could be extended to more complicated feature ensembles, but it would likely be quite involved.

As corroborated by our experiments in Figs. 2c and 2d, in practice one does not observe sharp transitions in test error from $\frac{1}{2}$ to $0$ using weak supervision — such a sharp transition corresponds to a Performance Gap Recovered (PGR) equal to 1 in Burns et al. (2023) This phenomenon can nevertheless be justified by our theory. First, the transition we prove is asymptotic; the rate of convergence matters to predict the PGR for finite $n$. Another factor is the 1-sparse assumption for the true labels. We argue in Appendix G that this is essentially necessary to get this sharp transition in asymptotic test error. In practice, the true directions are rarely aligned with the eigenvectors of the covariance, so the transition would instead between different *constant* levels of test error.

**Weak imitation.** Our main result Theorem 3.2 uses the subset ensemble Assumption 2 to establish two distinct phases of weak-to-strong generalization. By adding a third intermediate eigenvalue level — thus relaxing the bi-level ensemble to a *tri-level* ensemble — and changing the relationship between the strong and weak basis, we expect a third distinct regime of weak imitation to occur in an overparameterized setting. At a high level, this can happen if the weak classifier puts most of its mass in its own unfavored direction $v$, but in the strong feature space $v$ has an intermediate level of favoring. Given enough weakly labeled datapoints, the strong model will eventually learn to imitate the weak model. Nevertheless, it is still possible for weak-to-strong generalization to occur, with the regime for weak-to-strong generalization widening as the intermediate favoring level decreases.

**Multiclass classification.** By leveraging Theorem 3.1 and the techniques developed for binary and multilabel classification in the appendix, we expect our result to extend to the multiclass setting. The main technical challenge is controlling the expected weak-to-strong training behavior, which is significantly more complicated. However, multilabel weak supervision overcomes this challenge and is more tractable to analyze, as discussed after Theorem 3.5.

**Relevance to realistic settings.** One connection to more realistic settings comes through the NTK perspective, as touched upon in the introduction. To reiterate, if finetuning remains in the lazy training phase, then the supervised finetuning dynamics are captured by an appropriate generalized linear model defined by the NTK approximation. The strong and weak features are therefore learned from the pretraining phase. Another relevant setting is training linear probes on intermediate activations to interpret large models (Alain & Bengio, 2016; Marks & Tegmark, 2023; Nanda et al., 2023; Zou et al., 2023). Hence, under appropriate overparameterization of the intermediate activations, our results also apply there. To go beyond classification problems, one could hope to use some recent implicit bias results for next token prediction (Thrampoulidis, 2024) to study language models.

## ACKNOWLEDGMENTS

We would like to acknowledge support from an OpenAI Superalignment grant and NSF AST-2132700.

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

# A  PRELIMINARIES

In this section, we lay out some preliminaries for the rigorous justification of our main results, and give a technical overview in Section 3.1 which motivates the study of the survival and contamination of the signal, defined formally in Definition 3. In Appendix B, we will introduce tools from high dimensional probability which can rigorously control the survival and contamination. In Appendix C, we will then use these tools to prove the main results of the paper for binary classification. Then, in Appendix D, we will discuss how the same exact proofs extend to the multilabel setting, and in Appendix E we will prove a tight lower tail inequality for correlated Gaussians which yields Theorem 3.1. In Appendix F we include some simulations validating in our theory, and finally in Appendix G we give a heuristic calculation which arrives at the same predictions for the weak-to-strong regimes which should lead to rigorous proofs even without the 1-sparse assumption.

**Notation.** For positive integers $n$, we use the shorthand $[n] \triangleq \{1, \ldots, n\}$. For a vector $v \in \mathbb{R}^n$, $\|v\|_2$ always denotes the Euclidean norm. We index entries by using square brackets or subscripts, whichever is clearer, so $v[j]$ and $v_j$ both denote the $j$th entry of $v$. For any matrix $M \in \mathbb{R}^{m \times n}$, we denote its $ij$th entry by $m_{ij}$, $\|M\|_2$ denotes the spectral norm, and $\|M\|_F = \text{Tr}(M^\top M)$ denotes the Frobenius norm. If $M \in \mathbb{R}^{n \times n}$ is symmetric, we write $\mu_1(M) \geqslant \mu_2(M) \geqslant \ldots \geqslant \mu_n(M)$ to denote the ordered eigenvalues of $M$.

We make extensive use of asymptotic notation. We say $f(n) \asymp g(n)$ if $f(n) = \Theta(g(n))$, $f(n) \lesssim g(n)$ if $f(n) = O(g(n))$, and $f(n) = \widetilde{O}(g(n))$ if $f(n) = O(g(n) \log(n)^c)$ for some constant $c \geqslant 0$. When we write $f(n) \gg g(n)$, we mean that $\frac{f(n)}{g(n)} \geqslant n^\varepsilon$ for some constant $\varepsilon > 0$. An event $\mathcal{E}$ is said to hold with high probability if it holds with probability $1 - 1/n^c$ for some constant $c > 0$, and very high probability if it holds with probability $1 - \exp(-n^c)$ for some constant $c > 0$. For us, high probability statements will be taken with respect to $n$, the number of datapoints we scale the weak

and strong features with respect to, but under our terminology it makes no difference whether we use $n$ or $m$ because they are polynomially related.

In the weak-to-strong ensemble analysis, many of the expressions will depend on $q + r$, but the number of datapoints involved will change. In an effort to reduce confusion, we introduce the following notation.

**Definition 4** (Bi-level prefactor). *If the covariance $\Sigma = \Sigma(p, q, r)$ is drawn from the bi-level ensemble scaled with respect to $n$, then we define*

$$\mu_n \triangleq \frac{an}{s} = n^{1-q-r}. \tag{9}$$

*In particular, if $q + r > 1$ then $\mu_n \ll 1$. During weak supervision, the strong learner has bi-level features with covariance $\Sigma(p, q, r)$ scaled with respect to $n$ and receives $m = n^u$ weakly labeled examples. Hence, we define*

$$\mu_m \triangleq \frac{am}{s} = n^{u-q-r}. \tag{10}$$

**Remark A.1.** *Note that once $a, s$ are fixed, $\mu_n$ is linear in $n$.*

# B  TOOLS FOR ANALYZING SURVIVAL AND CONTAMINATION

In this section, we introduce the basic machinery which can justify the heuristic calculations carried out in Appendix A. We will heavily rely on the analysis of Wu & Sahai (2024), so in an attempt to not replicate effort we will comment on how the proofs change in the weak-to-strong setting.

**Basis change.**  Since we have bi-level data and the true direction $v_*$ is 1-sparse (Assumption 1), there are convenient basis changes which will greatly simplify the calculations. In particular, let $S$ be the spiked subspace of $\Sigma$, and $w$ be any unit vector. Then any vectors in $S$ are eigenvectors of $\Sigma$, so we can pick an eigenbasis $U_*$ so that $x_{\text{strong}}$ has independent coordinates, $v_*$ is 1-sparse, and $w$ is 3-sparse. We state this more formally in the following definition.

**Definition 5** (Strong eigenbasis). *Given $\Sigma$, $v_*$, and $w$ as above, let $U_*$ be the distinguished eigenbasis of $\Sigma$ such that after rotating to $U_*$, we have $x_{\text{strong}} \sim N(0, \Lambda)$, $v_* = e_1$, and $w = w_1 e_1 + w_2 e_2 + w_{s+1} e_{s+1}$.*

**Hanson Wright inequality.**  Now, we recall a few versions of the Hanson-Wright inequality which handles bilinear forms between subgaussian vectors.

For the multiclass case, we also need to handle potentially sparse vectors; this version was proved in Wu & Sahai (2024).

**Theorem B.1** (Hanson-Wright for bilinear forms with soft sparsity). *Let $x = (X_1, \ldots, X_n) \in \mathbb{R}^n$ and $y \in (Y_1, \ldots, Y_n) \in \mathbb{R}^n$ be random vectors such that $(X_i, Y_i)$ are independent pairs of (possibly correlated) centered random variables such that $\|X_i\|_{\psi_2} \leqslant K$ and $Y_i$ has soft sparsity at level $\sigma$, i.e. $|Y_i| \leqslant 1$ almost surely, and $\mathbb{E}[Y_i^2] \leqslant \sigma$. Assume that conditioned on $Y_j$, $\|X_j\|_{\psi_2} \leqslant K$. Then there exists an absolute constant $c > 0$ such that for all $M \in \mathbb{R}^{n \times n}$ and $\epsilon \geqslant 0$ we have*

$$\mathbf{Pr}\left[|x^\top M y - \mathbb{E}[x^\top M y]| > \epsilon\right] \leqslant 2 \exp\left(-c \min\left\{\frac{\epsilon^2}{K^2 \sigma \|M\|_F^2}, \frac{\epsilon}{K \|M\|_2}\right\}\right). \tag{11}$$

Because we are working with MNI, in a typical application we hope to set $M = A^{-1}$. However, in order to apply Hanson-Wright, one needs to ensure that the matrix $M$ is independent of $x$ and $y$.

**Decorrelated Gram matrix.**  As we are working with Gaussian covariates, one way to achieve this is to decorrelate $A$ with $x$ and $y$.

**Definition 6** (Decorrelated Gram matrix). *Let $x_1, \ldots, x_n \sim N(0, \Lambda)$ be iid samples from a Gaussian distribution in $\mathbb{R}^d$. Let $R = \{v^{(1)}, \ldots, v^{(\ell)}\}$ be an arbitrary set of orthonormal vectors in $\mathbb{R}^d$ and for each $j \in [\ell]$, let $h^{(j)} = (h_1, \ldots, h_n) \in \mathbb{R}^n$ be the corresponding Gaussian observations, where $h_i \sim \langle \Lambda^{1/2} g_i, v^{(j)} \rangle$.*

*Define the projection matrix onto $R$ by $\Pi_R = \sum_{\boldsymbol{v} \in R} \boldsymbol{v}\boldsymbol{v}^\top$. The $R$-projection of the data matrix $\boldsymbol{X}_R \in \mathbb{R}^{n \times d}$ is defined by*

$$\boldsymbol{X}_R \triangleq \boldsymbol{X}\Pi_R,$$

*and the $R$-decorrelated data matrix $\boldsymbol{X}_{-R} \in \mathbb{R}^{n \times d}$ is defined by*

$$\boldsymbol{X}_{-R} \triangleq \boldsymbol{X}(I - \Pi_R) = \boldsymbol{X} - \boldsymbol{X}_R.$$

*The decorrelated Gram matrix is defined by*

$$\boldsymbol{A}_{-R} \triangleq \boldsymbol{X}_{-R}\boldsymbol{X}^\top_{-R} = \boldsymbol{X}(I - \Pi_R)\boldsymbol{X}^\top.$$

*When $R = \{\boldsymbol{v}\}$, we will simply abbreviate $\{\boldsymbol{v}\}$ with $\boldsymbol{v}$.*

**Remark B.2.** *If $R$ does not consist of orthonormal vectors, one can apply Gram-Schmidt first so that the above definition still makes sense.*

It is not hard to show that as defined above, $\boldsymbol{A}_{-R}$ is mutually independent of $\left\{\boldsymbol{h}^{(j)}\right\}_{j \in [\ell]}$,

**Fact B.3.** *In the same setting as Definition 6, $\boldsymbol{A}_{-R}$ is mutually independent of $\left\{\boldsymbol{h}^{(j)}\right\}_{j \in [\ell]}$.*

*Proof.* The fact that $\boldsymbol{X}_{-R}$ is independent of the $h_i$'s is due to the fact that if $g, h$ are jointly Gaussian with zero mean and unit variance, then $g - \mathbf{E}[gh]h$ is independent of $h$. This immediately generalizes to each row of $\boldsymbol{X}$, since the entries are all independent for any row. Finally, as the rows are independent, the statement is proved. □

To relate a bilinear form against $\boldsymbol{A}_{-R}^{-1}$ to the original bilinear form against $\boldsymbol{A}^{-1}$, we will also need the Woodbury inversion formula for inverting the Gram matrix after decorrelating it with various features.

Define the hat matrix for $R$ by

$$\boldsymbol{H}_R \triangleq \boldsymbol{X}_R^\top \boldsymbol{A}_{-R}^{-1} \boldsymbol{X}_R. \tag{12}$$

These hat matrices appear in the Woodbury inversion formula. For the sake of notational compactness, define

$$\boldsymbol{M}_R \triangleq \boldsymbol{X}_R(I + \boldsymbol{H}_R)^{-1}\boldsymbol{X}_R^\top. \tag{13}$$

**Fact B.4** (Woodbury inversion formula). *We have*

$$\begin{aligned}
\boldsymbol{A}^{-1} &= (\boldsymbol{X}_R\boldsymbol{X}_R^\top + \boldsymbol{A}_{-R})^{-1} \\
&= \boldsymbol{A}_{-R}^{-1} - \boldsymbol{A}_{-R}^{-1}\boldsymbol{X}_R(I + \boldsymbol{H}_R)^{-1}\boldsymbol{X}_R^\top\boldsymbol{A}_{-R}^{-1} \tag{14} \\
&= \boldsymbol{A}_{-R}^{-1} - \boldsymbol{A}_{-R}^{-1}\boldsymbol{M}_R\boldsymbol{A}_{-R}^{-1}. \tag{15}
\end{aligned}$$

**Concentration of spectrum.** Finally, in view of Hanson-Wright, we will need to control $\left\|\boldsymbol{A}_{-R}^{-1}\right\|_2$ and $\mathrm{Tr}\left(\boldsymbol{A}_{-R}^{-1}\right)$. To this end, we control the spectrum of $\boldsymbol{A}_{-R}$. For any PSD matrix $\Sigma$ with eigenvalues $\Lambda = \mathrm{diag}(\lambda_i)_{i \in [d]}$ sorted in descending order and $k \in [d]$, define the effective rank $r_k(\Sigma) \triangleq \frac{\sum_{i \geqslant k} \lambda_i}{\lambda_k}$. We will use the following bounds adapted from Bartlett et al. (2020), which show that the effective rank of the covariance matrix controls the spectrum of the Gram matrix (and hence the sample covariance).

**Lemma B.5** (Eigenvalue bounds from Bartlett et al. (2020)). *Suppose that $\boldsymbol{x} \sim N(0, \Sigma)$, where $\Sigma = U\Lambda U^\top$. Write $\Lambda = \mathrm{diag}(\lambda_i)$, and let $\boldsymbol{A} = \boldsymbol{X}\boldsymbol{X}^\top \in \mathbb{R}^{n \times n}$ denote the Gram matrix for $n$ iid observations of $\boldsymbol{x}$. If $\boldsymbol{x} = U^\top\Lambda^{1/2}\boldsymbol{z}$, where $\boldsymbol{z}$ has independent, unit variance, $O(1)$-subgaussian coordinates, the following holds.*

*If $r_1(\Sigma) = \omega(n)$, then with very high probability*

$$\mathrm{Tr}(\Lambda)(1 - o(1)) \leqslant \mu_n(\boldsymbol{A}) \leqslant \mu_1(\boldsymbol{A}) \leqslant \mathrm{Tr}(\Lambda)(1 + o(1))$$

As a corollary, we show that in the regime $q + r > 1$ in the bi-level ensemble, decorrelating the Gram matrix with any sublinear number of jointly Gaussian random variables will yield a flat matrix.

**Corollary B.6.** *Suppose that $x \sim N(0, \Sigma)$, where $\Sigma \in \mathbb{R}^{d \times d}$ satisfies $\mu_1(\Sigma) = o(\frac{d}{n})$ and for any $\ell = o(d)$, we have $\mu_\ell(\Sigma) r_\ell(\Sigma) = d(1 - o(1))$ (in particular, this holds if $\Sigma$ follows the bi-level ensemble and $q + r > 1$). Let $R$ be an arbitrary set of orthonormal vectors in $\mathbb{R}^d$ such that $|R| = o(d)$, and let $\boldsymbol{A}_{-R}$ be the corresponding decorrelated Gram matrix for $n$ iid observations of $x$ as in Definition 6. Then with very high probability,*

$$d(1 - o_n(1)) \leqslant \mu_n(\boldsymbol{A}_{-R}) \leqslant \mu_1(\boldsymbol{A}_{-R}) \leqslant d(1 + o_n(1))$$

*Proof.* First, note that the rows of $\boldsymbol{X}_{-R}$ are iid samples from $N(0, \Sigma_{-R})$, where $\Sigma_{-R} = (I - \Pi_R)\Sigma(I - \Pi_R)$.

Let $\ell = |R|$; by assumption $\ell = o(d)$. Since $I - \Pi_R$ is a projection matrix, we conclude by Cauchy Interlacing that $\mu_1(\Sigma_{-R}) \leqslant \mu_1(\Sigma)$ and for $i \in \{\ell + 1, \ldots, d\}$, $\mu_i(\Sigma) \leqslant \mu_i(\Sigma_{-R})$. It follows that

$$
\begin{aligned}
r_1(\Sigma_{-R}) = \frac{\text{Tr}(\Sigma_{-R})}{\mu_1(\Sigma_{-R})} &\geqslant \frac{\mu_\ell(\Sigma) r_\ell(\Sigma)}{\mu_1(\Sigma)} \\
&\geqslant \omega\left(\frac{d(1 - o(1))}{d/n}\right) \quad\quad \text{(Assumption on } \Sigma) \\
&\geqslant \omega(n).
\end{aligned}
$$

Thus Lemma B.5 implies that $\mu_i(\boldsymbol{A}_{-R}) = d(1 \pm o(1))$ with high probability.

$\square$

This leads to the following bounds on the spectra of $\boldsymbol{A}_{-R}^{-1}$ and the hat matrix $\boldsymbol{H}_R$; this generalizes Wu & Sahai (2024, Corollary B.5, Proposition B.6).

**Lemma B.7.** *In the bi-level model, for any set of orthonormal vectors $R$ with $|R| = o(d)$, if $q + r > 1$, then with very high probability we have*

$$
\begin{aligned}
\text{Tr}\big(\boldsymbol{A}_{-R}^{-1}\big) &= n^{1-p}(1 \pm o(1)) \\
\big\|\boldsymbol{A}_{-R}^{-1}\big\|_2 &\leqslant (1 + o(1))n^{-p} \\
\big\|\boldsymbol{X}_R \boldsymbol{X}_R^\top\big\|_2 &\leqslant o(n^p) \\
\mu_i(\boldsymbol{H}_R) &= 1 \pm o(1), \quad i \in [d]
\end{aligned}
$$

*Proof.* The first two statements are an immediate consequence of Corollary B.6. For the third, let $S, T \subseteq \mathbb{R}^d$ be the spiked and unspiked subspaces of $\Sigma$. Since $\boldsymbol{X}_R = (\boldsymbol{X}_S + \boldsymbol{X}_T)\Pi_R$ and using $(A + B)M(A + B) \preceq 2AMA + 2BMB$, we have

$$\boldsymbol{X}_R \boldsymbol{X}_R^\top \preceq 2\boldsymbol{X}_S \boldsymbol{X}_S^\top + 2\boldsymbol{X}_T \Pi_R \boldsymbol{X}_T^\top.$$

Wu & Sahai (2024, Lemma I.1) implies that $\big\|\boldsymbol{X}_S \boldsymbol{X}_S^\top\big\|_2 \leqslant n^{1-q-r} \cdot n^p = o(n^p)$ since $q + r > 1$. On the other hand, since the law of $\boldsymbol{X}_T$ is rotationally invariant, one can view $\boldsymbol{X}_T \Pi_R \boldsymbol{X}_T^\top$ as a Wishart matrix $\boldsymbol{Y}\boldsymbol{Y}^\top$ where $\boldsymbol{Y} \in \mathbb{R}^{n \times |R|}$. Hence the concentration of singular values for a matrix with independent subgaussian rows (see, e.g. Vershynin (2010, Theorem 5.39)) implies that with probability $1 - e^{-\sqrt{n}}$,

$$\big\|\boldsymbol{X}_T \Pi_R \boldsymbol{X}_T^\top\big\|_2 \leqslant (\sqrt{|R|} + \sqrt{n} + n^{1/4})^2 = O(R + n) = o(n^p).$$

Combining this with the bound on $\big\|\boldsymbol{X}_S \boldsymbol{X}_S^\top\big\|_2$, we conclude that $\big\|\boldsymbol{X}_R \boldsymbol{X}_R^\top\big\|_2 = o(n^p)$.

For the last statement, since $\boldsymbol{H}_R = I + \boldsymbol{X}_R^\top \boldsymbol{A}_{-R}^{-1} \boldsymbol{X}_R$, by applying the earlier bounds that $\mu_i(\boldsymbol{A}_{-R}^{-1}) = n^p(1 \pm o(1))$ and $0 \preceq \boldsymbol{X}_R^\top \boldsymbol{X}_R \preceq o(n^p)$, we conclude that $\mu_i(\boldsymbol{H}_R) = 1 \pm o(1)$. $\square$

**Hanson-Wright calculations.** Combining the above results, we can establish a concentration inequality for the bilinear form $\boldsymbol{z}_i^\top \boldsymbol{A}^{-1} \boldsymbol{y}$.

**Proposition B.8.** *Let $y = \mathrm{sgn}(\langle g, w \rangle)$ for some unit vector $w \in \mathbb{R}^d$ and $R$ be any set of orthonormal vectors such that $\mathrm{span}(R) \supseteq \{e_i, w\}$ and $|R| = o(d)$. Then conditioned on $A_{-R}^{-1}$, with probability $1 - \frac{1}{n}$,*

$$\left| z_i^\top A_{-R}^{-1} y - \mathbf{E}[z_i^\top A_{-R}^{-1} y] \right| \leqslant c_1 \left\| A_{-R}^{-1} \right\|_F \sqrt{\log n},$$

*where $c_1$ is a positive constant. Furthermore, under the bi-level scaling, if $q + r > 1$, then with very high probability over $A_{-R}^{-1}$ we have*

$$\mathbf{E}[z_i^\top A_{-R}^{-1} y | A_{-R}^{-1}] = \left(\frac{2}{\pi}\right)^{3/2} (1 \pm o_n(1)) \arcsin(w_i) \cdot \frac{n}{d},$$

$$\left\| A_{-R}^{-1} \right\|_F = (1 \pm o_n(1)) \frac{\sqrt{n}}{d}.$$

*Proof.* Since $A_{-R}^{-1}$ is independent of $(z_i, y)$, we can apply Theorem B.1 to obtain the first statement.

For the other two expressions, we can apply Lemma B.7 since we assumed $q + r > 1$. We now compute the expectation using the noise stability formula:

$$\mathbf{E}[z_i^\top A_{-R}^{-1} y | A_{-R}^{-1}] = \mathrm{Tr}\left( A_{-R}^{-1} \mathbf{E}[y z_i^\top] \right)$$

$$= \left(\frac{2}{\pi}\right)^{3/2} \arcsin(w_i) \mathrm{Tr}\left( A_{-R}^{-1} \right) \qquad \text{(Noise stability)}$$

$$= \left(\frac{2}{\pi}\right)^{3/2} (1 \pm o_n(1)) \arcsin(w_i) \cdot \frac{n}{d}, \qquad \text{(Lemma B.7)}$$

where the last line holds with very high probability over $A_{-R}^{-1}$.

Finally, for the last statement we have

$$\left\| A_{-R}^{-1} \right\|_F = \sqrt{\sum_{i \in [n]} \mu_i (A_{-R}^{-1})^2}$$

$$= \sqrt{\sum_{i \in [n]} d^2 (1 \pm o(1))} \qquad \text{(Lemma B.7)}$$

$$= (1 \pm o(1)) \frac{\sqrt{n}}{d}.$$

$\square$

## B.1 CONTROLLING THE ERROR FROM WOODBURY

In the previous subsection, we introduced some tools for obtaining concentration of the coefficients of the MNI classifier $f_{\mathrm{strong}}$. The remaining survival and contamination reduce to understanding the typical behavior of bilinear forms $z_i^\top A^{-1} y$. To do so, we work in the distinguished basis $U_*$ introduced in Definition 5. We can thus pick $R = \{e_1, e_2, e_{s+1}\}$ and apply Hanson-Wright to the bilinear form $z_i^\top A_{-R}^{-1} y$. The next lemma bounds the error from replacing $A^{-1}$ with $A_{-R}^{-1}$.

**Lemma B.9.** *Suppose we are in the bi-level model, $y = \mathrm{sgn}(\langle g, w \rangle)$, and $q + r > 1$. Suppose in the basis $U_*$, $w = w_1 e_1 + w_2 e_2 + w_{s+1} e_{s+1}$. Let $R = \{e_1, e_2, e_{s+1}\}$. With probability $1 - O(1/n)$, we have uniformly over $i \in [d]$ that*

$$\left| z_i^\top A^{-1} y - z_i^\top A_{-R}^{-1} y \right| \lesssim \left\| z_i^\top A_{-R}^{-1} X_R \right\|_2 \left\| X_R^\top A_{-R}^{-1} y \right\|_2.$$

*Furthermore, in the bi-level scaling we have with probability $1 - O(1/n)$ that*

$$\left\| z_i^\top A_{-R}^{-1} X_R \right\|_2 = \begin{cases} \widetilde{O}(\sqrt{\mu_n} n^{\frac{1-p}{2}}) & i \in [2] \\ \widetilde{O}(\sqrt{\mu_n} n^{-\frac{p}{2}} + n^{1-p}) & i = s+1 \\ \widetilde{O}(\sqrt{\mu_n} n^{-\frac{p}{2}}) & \text{otherwise} \end{cases}$$

*and*

$$\left\| X_R^\top A_{-R}^{-1} y \right\|_2 \lesssim \sqrt{\mu_n}((|w_1| + |w_2|) n^{\frac{1-p}{2}} + \widetilde{O}(n^{-\frac{p}{2}})) + |w_{s+1}| n^{1-p} + \widetilde{O}(n^{\frac{1}{2}-p}).$$

*Proof.* By relating these two bilinear forms using Woodbury, we have

$$
\left| \boldsymbol{z}_i^\top \boldsymbol{A}^{-1} \boldsymbol{y} - \boldsymbol{z}_i^\top \boldsymbol{A}_{-R}^{-1} \boldsymbol{y} \right| = \left| \boldsymbol{z}_i^\top \boldsymbol{A}_{-R}^{-1} \boldsymbol{M}_R \boldsymbol{A}_{-R}^{-1} \boldsymbol{y} \right| \qquad \text{(Fact B.4)}
$$
$$
\leqslant \|\boldsymbol{H}_R\|_2 \left\| \boldsymbol{z}_i^\top \boldsymbol{A}_{-R}^{-1} \boldsymbol{X}_R \right\|_2 \left\| \boldsymbol{X}_R^\top \boldsymbol{A}_{-R}^{-1} \boldsymbol{y} \right\|_2 \qquad \text{(Cauchy-Schwarz)}
$$
$$
\leqslant (1 + o(1)) \left\| \boldsymbol{z}_i^\top \boldsymbol{A}_{-R}^{-1} \boldsymbol{X}_R \right\|_2 \left\| \boldsymbol{X}_R^\top \boldsymbol{A}_{-R}^{-1} \boldsymbol{y} \right\|_2 . \qquad \text{(Lemma B.7)}
$$

It remains to control the above norms, which we achieve through another application of Hanson-Wright. Indeed, we can compute

$$
\left\| \boldsymbol{z}_i^\top \boldsymbol{A}_{-R}^{-1} \boldsymbol{X}_R \right\|_2^2 = \sum_{j \in \{1, 2, s+1\}} \lambda_j (\boldsymbol{z}_i^\top \boldsymbol{A}_{-R}^{-1} \boldsymbol{z}_j)^2
$$

Now, by Hanson-Wright, we have with probability $1 - O(\frac{1}{n})$ that

$$
\lambda_U (\boldsymbol{z}_i^\top \boldsymbol{A}_{-R}^{-1} \boldsymbol{z}_{s+1})^2 = \begin{cases} \widetilde{O}(n^{2-2p}) & i = s+1 \\ \widetilde{O}(n^{1-2p}) & \text{otherwise} \end{cases}
$$

Similarly we have with probability $1 - O(\frac{1}{n})$ that

$$
\lambda_F \sum_{j \in [2]} (\boldsymbol{z}_i^\top \boldsymbol{A}_{-R}^{-1} \boldsymbol{z}_j)^2 = \begin{cases} \widetilde{O}(n^{p-q-r} \cdot n^{2-2p}) & i \in [2] \\ \widetilde{O}(n^{p-q-r} \cdot n^{1-2p}) & \text{otherwise} \end{cases}
$$

Since $n^{p-q-r} \gg 1$ as $p > q + r$ and $n^{p-q-r} \cdot n^{1-p} = \mu_n$, it follows that with probability $1 - O(\frac{1}{n})$ we have simultaneously for $i \in [d]$ that

$$
\left\| \boldsymbol{z}_i^\top \boldsymbol{A}_{-R}^{-1} \boldsymbol{X}_R \right\|_2 = \begin{cases} \widetilde{O}(\sqrt{\mu_n} n^{\frac{1-p}{2}}) & i \in [2] \\ \widetilde{O}(\sqrt{\mu_n} n^{-\frac{p}{2}} + n^{1-p}) & i = s+1 \\ \widetilde{O}(\sqrt{\mu_n} n^{-\frac{p}{2}}) & \text{otherwise} \end{cases}
$$

A similar calculation, applying Proposition B.8, yields that

$$
(\boldsymbol{z}_i^\top \boldsymbol{A}_{-R}^{-1} \boldsymbol{y})^2 = \begin{cases} \frac{1}{d^2} \left[ \left(\frac{2}{\pi}\right)^{3/2} \arcsin(w_i) n \pm \widetilde{O}(\sqrt{n}) \right]^2 & i \in \{1, 2, s+1\} \\ \widetilde{O}(\frac{n}{d^2}) & \text{otherwise} \end{cases}
$$

Hence,

$$
\left\| \boldsymbol{X}_R^\top \boldsymbol{A}_{-R}^{-1} \boldsymbol{y} \right\|_2 \lesssim \sum_{i \in \{1, 2, s+1\}} \frac{\sqrt{\lambda_i}}{d} (|w_i| n + \widetilde{O}(\sqrt{n}))
$$
$$
\lesssim \sqrt{\mu_n} (\|\boldsymbol{w}_S\|_1 n^{\frac{1-p}{2}} + \widetilde{O}(n^{-\frac{p}{2}})) + |w_{s+1}| n^{1-p} + \widetilde{O}(n^{\frac{1}{2}-p}).
$$

This concludes the proof. $\qquad \square$

We can simplify the above error bound greatly if $\boldsymbol{w}$ satisfies the 1-sparse assumption.

**Corollary B.10** (Woodbury error under 1-sparse assumption). *Suppose, in addition to the assumptions of Lemma B.9, we have $\boldsymbol{w} = \boldsymbol{v}_*$, where $\boldsymbol{v}_*$ satisfies the 1-sparse assumption. In other words, in the $\boldsymbol{U}_*$ basis we have $\boldsymbol{v}_* = \boldsymbol{e}_1$. Then we pick $R = \{\boldsymbol{e}_1\}$, and with probability $1 - O(\frac{1}{n})$ we have*

$$
\left| \boldsymbol{z}_i^\top \boldsymbol{A}^{-1} \boldsymbol{y} - \boldsymbol{z}_i^\top \boldsymbol{A}_{-R}^{-1} \boldsymbol{y} \right| \leqslant \begin{cases} \widetilde{O}(\mu_n \cdot n^{1-p}) & i = 1 \\ \widetilde{O}(\mu_n \cdot n^{\frac{1}{2}-p}) & \text{otherwise} \end{cases}
$$

## B.2 SURVIVAL ANALYSIS

Let us first establish a general survival bound using Lemma B.9. Afterward, we present special cases under additional assumptions.

**Proposition B.11.** *Let $\Sigma = \Sigma(p, q, r)$ be drawn from the bi-level ensemble scaled with respect to $n$ with $q + r > 1$. Suppose in the distinguished basis $U_*$, we have $\boldsymbol{w} = w_1\boldsymbol{e}_1 + w_2\boldsymbol{e}_2 + w_{s+1}\boldsymbol{e}_{s+1}$, and set $R = \{\boldsymbol{e}_1, \boldsymbol{e}_2, \boldsymbol{e}_{s+1}\}$. Then, given $n$ labels generated by $\boldsymbol{w}$, we have*

$$\left| \boldsymbol{z}_1^\top \boldsymbol{A}^{-1}\boldsymbol{y} - \left(\frac{2}{\pi}\right)^{3/2} \cdot \frac{n}{d}(1 + o(1))\arcsin w_1 \right| \lesssim \widetilde{O}\left(\frac{\sqrt{n}}{d}\right) + \left\| \boldsymbol{z}_i^\top \boldsymbol{A}_{-R}^{-1}\boldsymbol{X}_R \right\|_2 \left\| \boldsymbol{X}_R^\top \boldsymbol{A}_{-R}^{-1}\boldsymbol{y} \right\|_2.$$

*Proof.* Recall our expressions for the survival:

$$\mathsf{SU}_n(\boldsymbol{v}|\boldsymbol{w}) = \sum_{i \in [d]} \lambda_i \boldsymbol{z}_i^\top \boldsymbol{A}^{-1}\boldsymbol{y}\langle \boldsymbol{v}_i, \boldsymbol{v}\rangle, \qquad \text{(MNI)}$$

where each label is generated by $y_i = \text{sgn}(\langle \boldsymbol{g}_i, \boldsymbol{w}\rangle)$.

Setting $\boldsymbol{v} = \boldsymbol{v}_*$, in the basis $U_*$ from Definition 5, the above expression simplifies to

$$\mathsf{SU}_n(\boldsymbol{v}_*|\boldsymbol{w}) = \lambda_1 \boldsymbol{z}_1^\top \boldsymbol{A}^{-1}\boldsymbol{y}$$

Applying Woodbury (Fact B.4) with $R = \{\boldsymbol{v}, \boldsymbol{w}\}$, we have

$$\boldsymbol{z}_1^\top \boldsymbol{A}^{-1}\boldsymbol{y} = \boldsymbol{z}_1^\top \boldsymbol{A}_{-R}^{-1}\boldsymbol{y} - \boldsymbol{z}_1^\top \boldsymbol{A}_{-R}^{-1}\boldsymbol{M}_R\boldsymbol{A}_{-R}^{-1}\boldsymbol{y}.$$

Now we can apply Proposition B.8 to conclude that

$$\left| \boldsymbol{z}_1^\top \boldsymbol{A}_{-R}^{-1}\boldsymbol{y} - \left(\frac{2}{\pi}\right)^{3/2} \cdot \frac{n}{d}(1 + o(1))\arcsin(w_1) \right| \leqslant \widetilde{O}\left(\frac{\sqrt{n}}{d}\right)$$

with probability $1 - \frac{1}{n}$. Now, triangle inequality and Lemma B.9 shows that

$$\left| \boldsymbol{z}_1^\top \boldsymbol{A}^{-1}\boldsymbol{y} - \left(\frac{2}{\pi}\right)^{3/2} \cdot \frac{n}{d}(1 + o(1))\arcsin w_1 \right| \lesssim \widetilde{O}\left(\frac{\sqrt{n}}{d}\right) + \left\| \boldsymbol{z}_i^\top \boldsymbol{A}_{-R}^{-1}\boldsymbol{X}_R \right\|_2 \left\| \boldsymbol{X}_R^\top \boldsymbol{A}_{-R}^{-1}\boldsymbol{y} \right\|_2.$$

Thus the statement follows. $\qquad \square$

By way of Corollary B.10, we get a clean expression for the survival if $\boldsymbol{w} = \boldsymbol{v}_*$.

**Corollary B.12.** *If $q + r > 1$, $\boldsymbol{v}_*$ satisfies the 1-sparse assumption, and $\boldsymbol{y} = \text{sgn}(\langle \boldsymbol{g}, \boldsymbol{v}_*\rangle)$, then with probability $1 - O(1/n)$*

$$\boldsymbol{z}_1^\top \boldsymbol{A}^{-1}\boldsymbol{y} \asymp \frac{n}{d}$$

$$\left| \boldsymbol{z}_i^\top \boldsymbol{A}^{-1}\boldsymbol{y} \right| \leqslant \widetilde{O}\left(\frac{\sqrt{n}}{d}\right) \quad \forall i > 1$$

$$\mathsf{SU}_n(\boldsymbol{v}_*|\boldsymbol{v}_*) \asymp \mu_n$$

*Proof.* If $\boldsymbol{w} = \boldsymbol{v}_*$, then by Corollary B.10 the error bound simplifies to

$$\left| \boldsymbol{z}_i^\top \boldsymbol{A}^{-1}\boldsymbol{y} - \boldsymbol{z}_i^\top \boldsymbol{A}_{-R}^{-1}\boldsymbol{y} \right| \leqslant \begin{cases} \widetilde{O}(\mu_n \cdot \frac{n}{d}) & i = 1 \\ \widetilde{O}(\mu_n \cdot \frac{n}{d}) & \text{otherwise} \end{cases}$$

Since $q + r > 1$, these error terms are lower order, and the results follow. $\qquad \square$

We can get a similarly clean bound under a few additional assumptions on $\boldsymbol{w}$ and $q + r$.

**Corollary B.13.** *If, in addition to the assumptions of Proposition B.11, we have*

(1) $|w_i| \geqslant \omega(\sqrt{\frac{\log n}{n}})$ *for $i \in \{1, 2, s + 1\}$*

(2) $|w_2| = o(|w_1|)$

(3) $\sqrt{\mu_n} n^{\frac{1-p}{2}} \ll |w_1|$

*then with $y_i = \text{sgn}(\langle g_i, w \rangle)$, we have*

$$z_1^\top A^{-1} y \asymp \frac{n}{d} \arcsin w_1.$$

*Furthermore, the survival of $v_*$ given $n$ labels generated by $w$ is with high probability*

$$\mathsf{SU}_n(v_* | w) \asymp \mu_n w_1 \tag{16}$$

*Proof.* Under the additional assumptions, then from Lemma B.9

$$\left\| X_R^\top A_{-R}^{-1} y \right\|_2 \lesssim \sqrt{\mu_n}(|w_1| + |w_2|)n^{\frac{1-p}{2}} + n^{1-p} \qquad (|w_i| = \omega(\sqrt{\log n / n}))$$

$$\lesssim \sqrt{\mu_n}|w_1|n^{\frac{1-p}{2}} + n^{1-p} \qquad (|w_2| = o(|w_1|))$$

Since $i = 1$ here, we can apply Lemma B.9 to see that $\left\| z_1^\top A_{-R}^{-1} X_R \right\|_2 = \widetilde{O}(\sqrt{\mu_n} n^{\frac{1-p}{2}})$. Thus, the Woodbury error term is upper bounded by

$$\widetilde{O}(\sqrt{\mu_n} n^{\frac{1-p}{2}})(\sqrt{\mu_n}|w_1|n^{\frac{1-p}{2}} + n^{1-p}) \leqslant \widetilde{O}(\mu_n|w_1|n^{1-p} + \sqrt{\mu_n} n^{\frac{3-3p}{2}})$$

$$\lesssim \widetilde{O}(\mu_n|w_1|n^{1-p}) + o(n^{1-p}|w_1|)$$

$$\leqslant o(n^{1-p}|w_1|)$$

where the second line follows because we assumed $\sqrt{\mu_n} n^{\frac{1-p}{2}} \ll |w_1|$, and the last line follows because we assumed $q + r > 1$ so $\mu_n \ll 1$. Therefore, Proposition B.8 implies that

$$\left| z_1^\top A^{-1} y - \left( \frac{2}{\pi} \right)^{3/2} n^{1-p}(1 + o(1)) \arcsin w_1 \right|$$

$$\lesssim \widetilde{O}(n^{\frac{1}{2}-p}) + o(n^{1-p} \arcsin(|w_1|))$$

$$\leqslant o(n^{1-p} \arcsin w_1), \qquad (|w_1| \geqslant \omega(\sqrt{\tfrac{\log n}{n}}))$$

which recovers the stated result. $\qquad \square$

## B.3 CONTAMINATION ANALYSIS

We now move onto the contamination.

**Proposition B.14.** *Suppose $v_*$ satisfies the 1-sparse assumption, and in the distinguished basis $U_*$ we have $v_* = e_1$ and $w = w_1 e_1 + w_2 e_2 + w_{s+1} e_{s+1}$. Under the same assumptions as Corollary B.13, if there are $n$ datapoints, and $q + r > 1$, then*

$$\mathsf{CN}_n(v_* | w)^2 \lesssim o(\mu_n^2 |w_1|^2) + \mu_n^2 n^{r-1} \log(n)^2 + n^{1-p} \log(n)$$

$$\mathsf{CN}_n(v_* | w)^2 \gtrsim \mu_n^2 n^{r-1} + n^{1-p}.$$

*Furthermore, the lower bound holds even without the additional assumptions from Corollary B.13.*

*Proof.* Since $v_*$ is 1-sparse, we have

$$\mathsf{CN}(v_* | w)^2 = \sum_{i \in \{2, s+1\}} (\lambda_i z_i^\top A^{-1} y)^2 + \sum_{i \notin \{1, 2, s+1\}} (\lambda_i z_i^\top A^{-1} y)^2. \tag{17}$$

Since by definition $w_i = 0$ for $i \notin \{1, 2, s+1\}$, the second term in Equation (17) can be upper bounded up to constant factors using Wu & Sahai (2024, Proposition A.2) by

$$\mu_n^2 \frac{s}{n} \log(n)^2 + \frac{n}{d} \log(n) = \mu_n^2 n^{r-1} \log(n)^2 + n^{1-p} \log(n).$$

Now, set $R = \{e_1, e_2, e_{s+1}\}$. For the first term, we know from Lemma B.9 that

$$\left| z_i^\top A^{-1} y - z_i^\top A_{-R}^{-1} y \right| \lesssim \left\| z_i^\top A_{-R}^{-1} X_R \right\|_2 \left\| X_R^\top A_{-R}^{-1} y \right\|_2,$$

where

$$\big\| \boldsymbol{z}_i^\top \boldsymbol{A}_{-R}^{-1} \boldsymbol{X}_R \big\|_2 = \begin{cases} \widetilde{O}(\sqrt{\mu_n} n^{\frac{1-p}{2}}) & i = 2 \\ \widetilde{O}(\sqrt{\mu_n} n^{-\frac{p}{2}} + n^{1-p}) & i = s+1 \end{cases}$$

and from our bounds in Proposition B.11, we have

$$\big\| \boldsymbol{X}_R^\top \boldsymbol{A}_{-R}^{-1} \boldsymbol{y} \big\|_2 \lesssim \sqrt{\mu_n} |w_1| n^{\frac{1-p}{2}} + n^{1-p}.$$

Hence we have

$$\big\| \boldsymbol{z}_2^\top \boldsymbol{A}_{-R}^{-1} \boldsymbol{X}_R \big\|_2 \big\| \boldsymbol{X}_R^\top \boldsymbol{A}_{-R}^{-1} \boldsymbol{y} \big\|_2 \leqslant o(n^{1-p} |w_1|).$$

On the other hand, we have

$$\big\| \boldsymbol{z}_{s+1}^\top \boldsymbol{A}_{-R}^{-1} \boldsymbol{X}_R \big\|_2 \big\| \boldsymbol{X}_R^\top \boldsymbol{A}_{-R}^{-1} \boldsymbol{y} \big\|_2 \lesssim \mu_n |w_1| n^{\frac{1}{2}-p} + \sqrt{\mu_n} |w_1| n^{\frac{3-3p}{2}} + \sqrt{\mu_n} n^{1-\frac{3p}{2}} + n^{2-2p}$$
$$\lesssim o(n^{\frac{1}{2}-p}) + o(n^{1-p} |w_1|) + n^{2-2p},$$

where the last line follows from the assumptions and $p > 1$. Putting these together with Hanson-Wright, we have

$$\big| \boldsymbol{z}_2^\top \boldsymbol{A}^{-1} \boldsymbol{y} \big| \leqslant \widetilde{O}(n^{\frac{1}{2}-p}) + o(n^{1-p} |w_1|)$$
$$\big| \boldsymbol{z}_{s+1}^\top \boldsymbol{A}^{-1} \boldsymbol{y} \big| \leqslant \widetilde{O}(n^{\frac{1}{2}-p}) + o(n^{1-p} |w_1|) + n^{2-2p}$$

This yields an upper bound of

$$o(\mu_n^2 |w_1|^2) + \widetilde{O}(n^{1-2p}) + o(n^{2-2p} |w_1|^2) + n^{4-4p} \leqslant o(\mu_n^2 |w_1|^2) + n^{1-p},$$

as $p > 1$ and $|w_1| \leqslant 1$. This completes the proof of the upper bound.

For the lower bound, a careful inspection of the proof of Wu & Sahai (2024, Proposition A.2) reveals that, for $q + r > 1$, the lower bound holds for any multiclass problem with $t \geqslant 0$. This recovers the desired bound by simply lower bounding the second term in the expansion Equation (17), completing the proof.

$\square$

# C  ANALYZING THE SUBSET ENSEMBLE

In this section, we prove Theorem 3.2, which establishes weak-to-strong generalization in the simple weak-to-strong subset ensemble. We restate it below for convenience.

**Theorem 3.2** (Weak-to-strong generalization for subset ensemble)**.** *Consider the setup in Procedure 1 where the weak model $\boldsymbol{f}_{\mathsf{weak}}$ is MNI-trained on $n$ correctly labeled examples and the strong model $\boldsymbol{f}_{\mathsf{w2s}}$ is MNI-trained on $m = n^u$ weakly pseudolabeled examples, where $q + r > u$ and $q_{\mathsf{weak}} + r_{\mathsf{weak}} > 1$. In addition, we make the following data assumptions:*

(1) *The true binary labels are 1-sparse (Assumption 1) for the strong covariance.*

(2) *The weak and strong features follow the subset ensemble (Assumption 2) with bi-level eigenvalues $\Lambda(p, q, r)$ and $\Lambda(p_{\mathsf{weak}}, q_{\mathsf{weak}}, r_{\mathsf{weak}})$, respectively, scaled relative to $n$.*

(3) *There are not too many weakly labeled examples: $u < \frac{p + 1 + q + r - (q_{\mathsf{weak}} + r_{\mathsf{weak}})}{2}$.*

*Recall $\tau_{\mathsf{strong}} \triangleq p + 1 - 2(q + r)$. Then, the resulting test error for $\boldsymbol{f}_{\mathsf{w2s}}$ satisfies*

$$\mathbf{E}[\ell(\boldsymbol{f}_{\mathsf{w2s}})] = \begin{cases} o_n(1), & \text{if } u > q_{\mathsf{weak}} + r_{\mathsf{weak}} - \min\{1 - r, \tau_{\mathsf{strong}}\} \\ \frac{1}{2} - o_n(1), & \text{if } u < q_{\mathsf{weak}} + r_{\mathsf{weak}} - \min\{1 - r, \tau_{\mathsf{strong}}\}. \end{cases} \tag{5}$$

*Proof.* The key idea of the proof is that, despite the error rate of the weak classifier being $\frac{1}{2} - o(1)$, the weak classifier still has a noticeable amount of mass on the true label defining direction. So long as the error rate is not $\frac{1}{2} - O(\frac{1}{\sqrt{m}})$, which random guessing can achieve, the strong learner can

still pick up on the signal contained in the weak labels. To make this precise, we will control the coordinates of $\boldsymbol{f}_{\text{weak}}$ in the distinguished basis (Definition 5).

Quantitatively, for the weak learner $\boldsymbol{f}_{\text{weak}}$, since we have clean labels and $q_{\text{weak}} + r_{\text{weak}} > 1$, we can apply Corollary B.12. Let $\mu_{n,\text{weak}} \triangleq \frac{a_{\text{weak}} n}{s_{\text{weak}}}$ denote the bi-level prefactor for the weak model. Then the corollary shows that in the basis where $\boldsymbol{v}_* = \boldsymbol{e}_1$, we have with high probability

$$\boldsymbol{f}_{\text{weak}}[1] = \sqrt{\lambda_{F,\text{weak}}} \boldsymbol{z}_1^\top \boldsymbol{A}^{-1} \boldsymbol{y}$$

$$\asymp \sqrt{\frac{a_{\text{weak}} d_{\text{weak}}}{s_{\text{weak}}}} \frac{n}{d_{\text{weak}}}$$

$$= \sqrt{\mu_{n,\text{weak}}} n^{\frac{1-p_{\text{weak}}}{2}},$$

whereas for all other coordinates we have with high probability

$$|\boldsymbol{f}_{\text{weak}}[i]| \lesssim \begin{cases} \sqrt{\lambda_{F,\text{weak}}} \cdot \frac{\sqrt{n}}{d_{\text{weak}}} \sqrt{\log n} & i \in \{2, \ldots, s_{\text{weak}}\} \\ \sqrt{\lambda_{U,\text{weak}}} \cdot \frac{\sqrt{n}}{d_{\text{weak}}} \sqrt{\log n} & i \in \{s_{\text{weak}} + 1, \ldots, d_{\text{weak}}\} \end{cases}$$

$$\lesssim \begin{cases} \sqrt{\mu_{n,\text{weak}}} n^{-\frac{p_{\text{weak}}}{2}} \sqrt{\log n} & i \in \{2, \ldots, s_{\text{weak}}\} \\ n^{\frac{1}{2} - p_{\text{weak}}} \sqrt{\log n} & i \in \{s_{\text{weak}} + 1, \ldots, d_{\text{weak}}\} \end{cases}$$

Hence, to obtain $\boldsymbol{w}_{\text{weak}}$ in $U_*$, we first normalize $\boldsymbol{f}_{\text{weak}}$ and then do the basis change. Some calculation yields

$$\|\boldsymbol{f}_{\text{weak}}\|^2 \lesssim \mu_{n,\text{weak}} n^{1-p_{\text{weak}}} + n^{1-p_{\text{weak}}} \log n \cdot \frac{s_{\text{weak}} \lambda_{F,\text{weak}} + (d_{\text{weak}} - s_{\text{weak}}) \lambda_{U,\text{weak}}}{d_{\text{weak}}}$$

$$= (\mu_{n,\text{weak}} + \log n) n^{1-p_{\text{weak}}}$$

$$\lesssim n^{1-p_{\text{weak}}} \log n \qquad\qquad (q_{\text{weak}} + r_{\text{weak}} > 1)$$

where the second line follows because $s_{\text{weak}} \lambda_{F,\text{weak}} + (d_{\text{weak}} - s_{\text{weak}}) \lambda_{U,\text{weak}} = d_{\text{weak}}$. Similarly, we can compute a simple lower bound:

$$\|\boldsymbol{f}_{\text{weak}}\|^2 \geqslant \sum_{i > s+1} (\boldsymbol{z}_i^\top \boldsymbol{A}^{-1} \boldsymbol{y})^2 = \boldsymbol{y}^\top \boldsymbol{A}^{-1} \sum_{i > s+1} \boldsymbol{z}_i \boldsymbol{z}_i^\top \boldsymbol{A}^{-1} \boldsymbol{y}$$

$$\gtrsim n^{1-p_{\text{weak}}}. \qquad\qquad \text{(Lemma B.7)}$$

We therefore obtain the sandwiching bounds

$$n^{\frac{1-p_{\text{weak}}}{2}} \lesssim \|\boldsymbol{f}_{\text{weak}}\| \lesssim n^{\frac{1-p_{\text{weak}}}{2}} \sqrt{\log n}. \tag{18}$$

By rotating to the distinguished basis, since $q_{\text{weak}} + r_{\text{weak}} > 1$, this yields

$$\boldsymbol{w}_{\text{weak}} = \frac{\boldsymbol{f}_{\text{weak}}}{\|\boldsymbol{f}_{\text{weak}}\|} = w_1 \boldsymbol{e}_1 + w_2 \boldsymbol{e}_2 + w_{s+1} \boldsymbol{e}_{s+1},$$

where

$$\frac{1}{\sqrt{\log n}} \sqrt{\mu_{n,\text{weak}}} \lesssim w_1 \lesssim \sqrt{\mu_{n,\text{weak}}} \tag{19}$$

$$\frac{1}{\log n} n^{-\frac{q_{\text{weak}}}{2}} \lesssim |w_2| \lesssim n^{-\frac{q_{\text{weak}}}{2}} \sqrt{\log n} \tag{20}$$

$$1 \lesssim |w_{s+1}|. \tag{21}$$

Let us now check the conditions to apply Corollary B.13, keeping in mind we need to scale with respect to $m$ instead of $n$. In particular, we require

- $w_i = \omega(\sqrt{\frac{\log m}{m}})$ for all $i \in \{1, 2, s+1\}$.

- $|w_2| \ll w_1$

- $\sqrt{\mu_m} n^{\frac{u-p}{2}} \ll |w_1|$

Since we are going to check the second condition, and clearly the first condition is satisfied for $i = s + 1$, it suffices to check the first condition for $i = 1$. By plugging in the scaling for $w_1$ in Equation (19), we have $w_1 = \omega(\sqrt{\frac{\log m}{m}})$ if

$$1 - q_{\text{weak}} - r_{\text{weak}} > -u \iff q_{\text{weak}} + r_{\text{weak}} < u + 1. \tag{22}$$

Let us now verify the second condition. It turns out that the second condition always holds under the bi-level parameterization. Indeed, from Equations (19) and (20), we always have

$$|w_2| \leqslant n^{-\frac{q_{\text{weak}}}{2}} \sqrt{\log n} \ll \frac{1}{\sqrt{\log n}} n^{\frac{1 - q_{\text{weak}} - r_{\text{weak}}}{2}} \leqslant w_1,$$

as $r_{\text{weak}} < 1$. Finally, for the third condition, we have

$$\sqrt{\mu_m} n^{\frac{u-p}{2}} = n^{\frac{u-q-r}{2} + \frac{u-p}{2}} \ll n^{\frac{1 - q_{\text{weak}} - r_{\text{weak}}}{2}} \iff 2u - (p + q + r) < 1 - (q_{\text{weak}} + r_{\text{weak}})$$
$$\iff q + r > 2u - (p + 1) + q_{\text{weak}} + r_{\text{weak}} \tag{23}$$

With these preparations in hand, we now prove the positive side of the result.

**Sufficient condition for weak-to-strong generalization.** First we lower bound $\mathsf{SU}_m(\boldsymbol{v}_* | \boldsymbol{w}_{\text{weak}})$. Under Equations (22) and (23), we conclude that

$$\mathsf{SU}_{m,\text{strong}}(\boldsymbol{v}_* | \boldsymbol{w}_{\text{weak}}) \gtrsim \mu_m w_1 \geqslant \frac{1}{\sqrt{\log n}} \mu_m \cdot n^{\frac{1 - q_{\text{weak}} - r_{\text{weak}}}{2}}$$

On the other hand, by Proposition B.14 we have

$$\mathsf{CN}_{m,\text{strong}}(\boldsymbol{v}_* | \boldsymbol{w})^2 \leqslant o(\mu_m^2 |w_1|^2) + \mu_m^2 n^{r-u} \log n + n^{u-p} \log n.$$

We will now analyze the survival-to-contamination ratio for the strong learner. The first term is clearly at least $\omega(1)$. For the second term, we get

$$\frac{1}{\log n} n^{\frac{1 - q_{\text{weak}} - r_{\text{weak}}}{2} - \frac{r-u}{2}},$$

which is $\omega(1)$ if

$$q_{\text{weak}} + r_{\text{weak}} < u + 1 - r \tag{24}$$

Finally, for the third term, we get

$$\frac{1}{\log n} n^{u-q-r + \frac{1 - q_{\text{weak}} - r_{\text{weak}}}{2} - \frac{u-p}{2}},$$

which is $\omega(1)$ if

$$q_{\text{weak}} + r_{\text{weak}} < u + p + 1 - 2(q + r). \tag{25}$$

Collecting the conditions Equations (22), (24) and (25) yields

$$u > q_{\text{weak}} + r_{\text{weak}} - \min\{1, 1 - r, p + 1 - 2(q + r)\}$$
$$= q_{\text{weak}} + r_{\text{weak}} - \min\{1 - r, p + 1 - 2(q + r)\},$$

which establishes the positive side of the result.

**Sufficient condition for failure of weak-to-strong generalization.** To get the negative result, we need to upper bound $\mathsf{SU}_{m,\text{strong}}(\boldsymbol{v}_* | \boldsymbol{w}_{\text{weak}})$ and lower bound $\mathsf{CN}_{m,\text{strong}}(\boldsymbol{v}_* | \boldsymbol{w}_{\text{weak}})$. Our earlier calculations and Proposition B.14 yield

$$|\mathsf{SU}_{m,\text{strong}}(\boldsymbol{v}_* | \boldsymbol{w}_{\text{weak}})| \lesssim \mu_m \cdot \max\left\{w_1, n^{\frac{u}{2} - p} \sqrt{\log n}\right\} \tag{26}$$

$$\mathsf{CN}_{m,\text{strong}}(\boldsymbol{v}_* | \boldsymbol{w}_{\text{weak}}) \gtrsim \mu_m n^{\frac{r-u}{2}} + n^{\frac{u-p}{2}}. \tag{27}$$

The above bound only differ by log factors from the upper bounds. Hence, in this case, the sufficient condition merely flips the direction of the inequality, which recovers the stated result. Otherwise, if $u < q_{\text{weak}} + r_{\text{weak}} - 1$, then certainly $u < q_{\text{weak}} + r_{\text{weak}} - 1 + r$, and $n^{\frac{u}{2} - p} \sqrt{\log n}$ dominates. The latter is in turn dominated by $n^{\frac{u-p}{2}}$, so the survival-to-contamination ratio is $o(1)$. This completes the proof. $\qquad\square$

## D  MULTILABEL CLASSIFICATION

In this section, we analyze multilabel classification with $k$ classes, which is a variant of multiclass classification where a single datapoint can be positive examples of multiple classes. Given $n$ datapoints, we encode this label enformation using $k$ label vectors $\boldsymbol{y}^{(1)}, \ldots, \boldsymbol{y}^{(k)}$, where for $i \in [k]$, we encode the $i$th label vector as $\boldsymbol{y}^{(i)} \in \{\pm 1\}^n$, with 1 representing positive examples. We make the following 1-sparse assumption for multilabel data.

**Assumption 3** (1-sparse assumption for multilabel). *The label defining directions $\boldsymbol{v}_*^{(1)}, \ldots, \boldsymbol{v}_*^{(k)}$ are aligned with a top $k$ eigenbasis of the strong covariance $\Sigma$. In a strong eigenbasis where $\boldsymbol{v}_*^{(i)} = \boldsymbol{e}_i$, we have*

$$y^{(j)} = \operatorname{sgn}(x_j), \qquad \forall j \in [k]$$

**Remark D.1.** *In fact, the analysis below will turn out to work even if the label defining directions are not orthogonal; they could all be the same! The analysis only relies on there existing some basis $U_i$ such that $\boldsymbol{v}_*^{(i)} = \boldsymbol{e}_1$. However, the tight misclassification rate would be different depending on the relationship between the label defining directions.*

The weak-to-strong setup is defined as follows.

(1)  $\boldsymbol{f}_{\mathsf{weak}} \in \mathbb{R}^{d_{\mathsf{weak}} \times k}$: train on $n$ datapoints using weak features and ground-truth labels.

(2)  $\boldsymbol{f}_{\mathsf{w2s}} \in \mathbb{R}^{d \times k}$: train on $m \gg n$ datapoints using strong features and hard pseudolabels generated from $\boldsymbol{f}_{\mathsf{weak}}$.

As before, we will study weak-to-strong generalization using overparameterized linear classifiers. Hence, $\boldsymbol{f}_{\mathsf{weak}}$ will now consist of $k$ different linear classifiers $\boldsymbol{f}_{\mathsf{weak}}^{(i)} \in \mathbb{R}^{d_{\mathsf{weak}}}$ for $i \in [k]$, all trained by MNI on $n$ clean multilabel datapoints. To train $\boldsymbol{f}_{\mathsf{w2s}}^{(i)}$, we generate hard pseudolabel vectors $\widehat{\boldsymbol{y}}^{(i)} = \operatorname{sgn}(\langle \boldsymbol{f}_{\mathsf{weak}}^{(i)}, \boldsymbol{x}_{\mathsf{weak}} \rangle)$, and then perform MNI on these hard pseudolabel vectors. We deem that a multilabel classifier $\boldsymbol{f}$ generalizes if, for a fresh test sample $\boldsymbol{x}_{\mathsf{test}}$ and for every class $i \in [k]$, $\boldsymbol{f}$ correctly labels whether $\boldsymbol{x}_{\mathsf{test}}$ is a positive example of class $i$. More formally, define for a collection of classifiers $\boldsymbol{f} = (\boldsymbol{f}^{(1)}, \ldots, \boldsymbol{f}^{(k)})$, the loss function $\ell(\boldsymbol{f}) = \mathbf{Pr}[\exists i \in [k] : \operatorname{sgn}(\boldsymbol{f}^{(i)}(\boldsymbol{x}_{\mathsf{test}})) \neq y_{\mathsf{test}}^{(i)}]$, where the probability is taken over a fresh test sample $(\boldsymbol{x}_{\mathsf{test}}, y_{\mathsf{test}})$.

**Remark D.2.** *The weak model can also be trained on clean multiclass data rather than clean multilabel data. This will only affect the regimes for the weak model to satisfy Desideratum 1.iii, based on Theorem 3.1.*

The subset ensemble definition is essentially the same as before (Assumption 2), with the main difference being that we require all $k$ of the label defining directions to be favored and axis-alignable in both the weak and strong favored feature subspace.

**Assumption 4** (Subset ensemble for multilabel classification). *Let $\Lambda = \Lambda(p, q, r) \in \mathbb{R}^{d \times d}$ denote the strong eigenvalues and $\Lambda_{\mathsf{weak}} = \Lambda(p_{\mathsf{weak}}, q_{\mathsf{weak}}, r_{\mathsf{weak}}) \in \mathbb{R}^{d_{\mathsf{weak}} \times d_{\mathsf{weak}}}$ denote the weak eigenvalues, both drawn from the bi-level ensemble. Suppose the 1-sparse assumption (Assumption 1) holds for the strong covariance, with any distinguished eigenbasis $U$ where $\boldsymbol{v}_*^{(i)} = \boldsymbol{e}_i$ for $i \in [k]$. The following conditions relate the weak and strong features after rotating to $U$.*

(1)  $\boldsymbol{x}_{\mathsf{strong}} \sim N(0, \Lambda)$, *where $\Lambda = \lambda_F I_{[s]} + \lambda_U I_{[d] \setminus [s]}$.*

(2)  *There exists subsets of coordinates $S \subseteq [s], T \subseteq [d] \setminus [s]$, with $[k] \subseteq S$ and $|S| = s_{\mathsf{weak}}$, such that*

$$\boldsymbol{x}_{\mathsf{weak}} = \left(\sqrt{\tfrac{\lambda_{F,\mathsf{weak}}}{\lambda_F}} \Pi_S + \sqrt{\tfrac{\lambda_{U,\mathsf{weak}}}{\lambda_U}} \Pi_T\right) \boldsymbol{x}_{\mathsf{strong}} \overset{d}{=} N(0, \lambda_{F,\mathsf{weak}} I_S + \lambda_{U,\mathsf{weak}} I_T).$$

The crucial observation is that multilabel training boils down to $k$ (nearly) independent binary classification problems. The main difference to establish successful generalization is that we now need to union bound over all $k$ multilabel classifiers. The high probability bounds from the binary analysis come from two sources: (1) applying spectral bounds (Lemma B.5), which holds with very high probability $(\exp(-n^{1/2}))$, and (2) Hanson-Wright calculations (Proposition B.8), where bounds

that hold with probability $\delta$ have deviation poly $\log(1/\delta)$. Hence, for $k$ a constant, $\delta$ is only affected by a constant, and this will only change the bounds in our analysis by a constant, which does not shift the regimes. Furthermore, even if we allow $k$ to scale with $n$ as in Definition 2, since $k$ is polynomial in $n$, the dependence for the high probability bounds on $k$ is at most polylogarithmic in $k$, so again nothing changes with the analysis, which looks at polynomial regime shifts.

For the converse direction, one can use a crude bound of just analyzing the probability of one classifier failing, which gives a error rate bounded from below by $\frac{1}{2} - o_n(1)$. However, we expect that a more refined analysis would give the expected error rate of $1 - O(2^{-k})$; we sketch an argument for how to get this improved error rate after the theorem statement. With these minor modifications, the formal details go through unchanged, and we arrive at our main theorem for weak-to-strong multilabel classification using hard pseudo-multilabels.

**Theorem D.3** (Weak-to-strong generalization for multilabel subset ensemble). *Consider the setup where the weak model $\boldsymbol{f}_{\mathsf{weak}}$ is trained on $n$ correctly labeled examples and the strong model $\boldsymbol{f}_{\mathsf{w2s}}$ is trained on $m = n^u$ weakly labeled examples using MNI, where $u < p$, $q + r > u$, and $q_{\mathsf{weak}} + r_{\mathsf{weak}} > 1$. Assume the following:*

(1) *The true multilabels satisfy the 1-sparse assumption (Assumption 3) for the strong covariance.*

(2) *The weak and strong features follow the subset ensemble (Assumption 2) with bi-level eigenvalues $\Lambda(p, q, r)$ and $\Lambda(p_{\mathsf{weak}}, q_{\mathsf{weak}}, r_{\mathsf{weak}})$, respectively, scaled relative to $n$.*

(3) *There are not too many weakly labeled examples: $u < \frac{p+1+q+r-(q_{\mathsf{weak}}+r_{\mathsf{weak}})}{2}$.*

*Let $\tau_{\mathsf{strong}} \triangleq p + 1 - 2(q + r)$. Then, the resulting test error for $\boldsymbol{f}_{\mathsf{w2s}}$ satisfies*

$$\mathbf{E}[\ell(\boldsymbol{f}_{\mathsf{w2s}})] = \begin{cases} o_n(1), & \text{if } u > q_{\mathsf{weak}} + r_{\mathsf{weak}} - \min\{1 - r, \tau_{\mathsf{strong}}\} \\ \Omega(1), & \text{if } u < q_{\mathsf{weak}} + r_{\mathsf{weak}} - \min\{1 - r, \tau_{\mathsf{strong}}\}. \end{cases} \tag{28}$$

We now sketch out an approach which should give the correct error rate for constant $k$ (or even $k$ growing slowly with $n$). Note that the complement of the failure event is

$$\forall i \in [k] : \mathrm{sgn}(\langle \boldsymbol{f}_{\mathsf{w2s}}^{(i)}, \boldsymbol{x}_{\mathsf{test}} \rangle) = y_{\mathsf{test}}^{(i)}.$$

From the Gram-Schmidt decomposition Equation (8) and the noise stability formula, we can decompose this event as

$$\forall i \in [k] : \mathrm{sgn}(\tfrac{\mathsf{SU}(\boldsymbol{v}_*^{(i)}|\boldsymbol{w})}{\mathsf{CN}(\boldsymbol{v}_*^{(i)}|\boldsymbol{w})} x_*^{(i)} + g^{(i)}) = \mathrm{sgn}(x_*^{(i)}),$$

where $x_*^{(i)} = \left\langle \boldsymbol{g}_{\mathsf{test}}, \boldsymbol{v}_*^{(i)} \right\rangle$ and $g^{(i)}$ are iid standard Gaussians. In the converse regime, the survival to contamination ratio is polynomially decaying, so for typical $g^{(i)}$, with probability $\exp(-n^c)$ we have $\mathrm{sgn}(\tfrac{\mathsf{SU}(\boldsymbol{v}_*^{(i)}|\boldsymbol{w})}{\mathsf{CN}(\boldsymbol{v}_*^{(i)}|\boldsymbol{w})} x_*^{(i)} + g^{(i)}) = \mathrm{sgn}(g^{(i)})$. Furthermore, by unpacking on the analysis of $\boldsymbol{f}_{\mathsf{w2s}}^{(i)}$ (specifically, see Equation (20)) for $i \neq j$, $g^{(i)}$ and $g^{(j)}$ only differ by a Gaussian with polynomially decaying variance, so up to a failure event with probability $\exp(-n^c)$, we can replace these with the same Gaussian $\widetilde{g}$ which is independent of both $x_*^{(i)}$ and $x_*^{(j)}$. We can do this for all pairs $(i, j)$ and union bound over $k$ (this is where we use the slow growing condition on $k$). Hence, up to these error terms, which are negligible, the complementary event occurs with probability $2^{-k}$, so the test error will indeed be $1 - O(2^{-k})$.

## D.1 MULTILABEL SUPERVISION FOR MULTICLASS CLASSIFICATION

In this section, we expand upon the arguments to use weak multilabel supervision to train a weak-to-strong multiclass classifier $\boldsymbol{f}_{\mathsf{w2s}}$. The difficulty of the multiclass analysis is studying the MNI behavior for weak supervision. However, since the multilabel MNI behavior is well under control by the arguments above, it is tractable to study this setup instead.

The key insight is that, for multiclass classification to succeed, it suffices to look at pairwise comparisons between the score functions for the $k$ different classes (see Wu & Sahai (2024) for more

justification). In particular, one studies the relative survival given different class label vectors:

$$\mathsf{SU}(\boldsymbol{v}|\widehat{\boldsymbol{y}}_{\mathsf{weak}}^{(i)}) - \mathsf{SU}(\boldsymbol{v}|\widehat{\boldsymbol{y}}_{\mathsf{weak}}^{(j)}),$$

with the signal component for class $i$ being the above quantity with $\boldsymbol{v} = \boldsymbol{v}_*^{(i)}$. One can then define the relative contamination in analogous way. However, the path towards studying the survival comes from understanding the coefficients of $\boldsymbol{f}_{\mathsf{weak}}$ and $\boldsymbol{f}_{\mathsf{w2s}}$, which is feasible.

Thus, to determine whether $\boldsymbol{f}_{\mathsf{w2s}}$ generalizes for multiclass classification, it reduces back down to the multilabel/binary behavior. As argued in Wu & Sahai (2024), because of the margin between the features and the survivals, it suffices to get a polynomially increasing $\mathsf{SU}/\mathsf{CN}$ ratio. Hence, in the successful regime for weak-to-strong multilabel generalization, this relative $\mathsf{SU}/\mathsf{CN}$ ratio is polynomially increasing, which implies that multiclass classification succeeds.

## E    IMPROVING THE BOUNDS FOR MISCLASSIFICATION RATE

In this section we tighten the bounds on misclassification rate for the multiclass setting.

**Theorem E.1** (Tightening of (Wu & Sahai, 2024, Proposition A.7))**.** *Assume we are in the bi-level ensemble model (Definition 1), the true data generating process is 1-sparse (Assumption 1), and the number of classes follows the scaling defined in Definition 2. Then, in the negative regime where the model does not achieve vanishing error, we have*

$$\mathbf{E}[\ell(\boldsymbol{f}_{\mathsf{strong}})] = 1 - \Theta\left(\frac{1}{k}\right), \tag{29}$$

*where the expectation is taken over the randomness of the training data and the test point.*

At a high level, the misclassification event $\mathcal{E}_{\mathsf{err}}$ is governed by the following inequality holding:

$$\frac{\mathsf{SU}_n}{\mathsf{CN}_n} \max_{j \in [k]} |x_j| \leqslant \max_{i \in [k]} g_i,$$

where $(x_j)_{j \in [k]}$ are iid standard normals, and $(g_i)_{i \in [k]}$ are jointly gaussian with some correlation structure.

Now, (Wu & Sahai, 2024, Proposition A.3) implies that $\frac{\mathsf{SU}_n}{\mathsf{CN}_n} \leqslant n^{-u}$ for some constant $u > 0$ with probability $1 - O(1/n)$. Also, we can upper bound $\max_{j \in [k]} |x_j| \leqslant O(\sqrt{\log k})$ with probability at least $1 - O(1/k)$. Moreover, in the regime where classification fails, we know that from (Wu & Sahai, 2024, Proposition F.1), we have that $\mathbf{E}[g_i g_j] \leqslant \frac{1}{2} + n^{-\delta}$ for some constant $\delta > 0$ with probability at least $1 - O(1/n)$. Since $k = o(n)$, when we union bound over the above events, they get absorbed by the $O(\cdot)$ and $\Omega(\cdot)$ terms.

To tighten the misclassification rate, we will improve upon (Lopes & Yao, 2022, Theorem 2.1). In particular, we shows the following lower tail inequality for the maximum of correlated gaussians. We will prove the following theorem, which captures the lower tail behavior of the maximum of correalted Gaussians very far from its expectation. This result complements the lower tail bound of Lopes & Yao (2022), which holds for more moderate deviations.

**Theorem 3.6** (Lower tail for correlated Gaussians)**.** *Let $\rho_0 \in (0, 1)$ be a parameter bounded away from 0 and 1, and let $(g_i)_{i \in [N]}$ be jointly Gaussian with zero mean and unit variance. Suppose $\mathbf{E}[g_i g_j] \leqslant \rho_0$ for all distinct $i, j \in [N]$. For any $0 \leqslant t_N = \delta_0 \sqrt{2(1 - \rho_0) \log N}$ where $\delta_0 \in [0, 1)$ is bounded away from 1, there is a constant $C > 0$ depending only on $\rho_0$ such that*

$$\mathbf{Pr}\left[\max_{i \in [N]} g_i \leqslant t_N\right] \leqslant C \cdot N^{(1-\delta_0)^2(1-\frac{1}{\rho_0})}(\log N)^{\frac{1-\rho_0(2-\delta_0)-\delta_0}{2\rho_0}}.$$

*In particular, one can take $C = \sqrt{\frac{\rho_0}{1-\rho_0}}$. If we further have $\mathbf{E}[g_i g_j] = \rho_0$ for all distinct $i, j$ and $t_N = O\left(\frac{\log \log N}{\sqrt{\log N}}\right)$, then $\mathbf{Pr}\left[\max_{i \in [N]} g_i \leqslant t_N\right] = \Theta\left(N^{1-\frac{1}{\rho_0}}(\log N)^{\frac{1}{2\rho_0}-1}\right)$.*

Before we prove the theorem, let us see how it tightens the misclassiifcation rate.

*Proof of Theorem E.1.* We will apply Theorem 3.6 with $N = k - 1$, $\rho_0 = \frac{1}{2} + n^{-\delta}$, and $\delta_0 = \frac{1}{\log k}$.

Noting that $\delta_0 \sqrt{2(1 - \rho_0) \log k} \geqslant n^{-u}$ if $k = c_k n^t$ for any $t \in [0, 1)$, we see that

$$\mathbf{Pr}[\max_{i \in [k]} g_i \leqslant n^{-u}] \leqslant \mathbf{Pr}[\max_{i \in [k]} g_i \leqslant \delta_0 \sqrt{2(1 - \rho_0) \log k}] \qquad (\delta_0 \sqrt{2(1 - \rho_0) \log k} \geqslant n^{-u})$$

$$\leqslant O\left(k^{-(1-\delta_0)^2(1+n^{-\delta})} (\log k)^{n^{-\delta} - \frac{\delta_0}{2}}\right) \qquad \text{(Theorem 3.6)}$$

$$\leqslant O\left(k^{-(1-\delta_0)^2} (\log k)^{-\frac{\delta_0}{2}}\right)(1 + o_n(1)) \qquad (k^{n^{-\delta}} = 1 + o_n(1))$$

$$\leqslant O(k^{-(1-\delta_0)^2}) \qquad ((\log k)^{-\delta_0/2} = 1 - o_k(1))$$

$$\leqslant O\left(\frac{1}{k}\right). \qquad (\delta_0 = 1/\log k)$$

Inverting this bound, we see that

$$\mathbf{Pr}[\max_{i \in [k]} g_i > n^{-u}] \geqslant 1 - O\left(\frac{1}{k}\right).$$

For the upper bound on the above probability, we can use Slepian's lemma and then compare to Gaussians $\underline{g_i}$ which have correlation $\frac{1}{2} - n^{-\delta}$. Writing it all out, we have

$$\mathbf{Pr}[\max_{i \in [k]} g_i \leqslant n^{-u}] \geqslant \mathbf{Pr}[\max_{i \in [k]} \underline{g_i} \leqslant n^{-u}] \qquad \text{(Slepian's lemma)}$$

$$\geqslant \mathbf{Pr}[\max_{i \in [k]} \underline{g_i} \leqslant 0]$$

$$\geqslant \Omega\left(\frac{1}{k^{1+n^{-\delta}}}\right) \qquad ((\text{Pinasco et al., 2021, Theorem 2.1}))$$

$$\geqslant \Omega\left(\frac{1}{k}\right). \qquad (k^{n^{-\delta}} = 1 + o_n(1))$$

from which we get a nearly matching upper bound on the probability of $1 - \Omega(\frac{1}{k})$. $\qquad \square$

We return to the proof of Theorem 3.6.

*Proof of Theorem 3.6.* The lower bound directly follows from the proof of Lopes & Yao (2022), so we focus on proving the upper bound. We remark that the constant hidden by $\Theta$ can be pinpointed to $(1 + o_N(1))\sqrt{\frac{\rho_0}{1-\rho_0}}$, but we will not discuss this further.

To reduce confusion, we will attempt to follow the notation and treatment from Lopes & Yao (2022). To prove the upper bound, we can use Slepian's lemma to reduce to the case where $\mathbf{E}[g_i g_j] = \rho_0$ for all $i \neq j$. Then, we can explicitly decompose

$$g_i = \sqrt{\rho_0} x + \sqrt{1 - \rho_0} h_i,$$

where $x, h_i$ are iid standard Gaussians. Via this decomposition, we can write an integral representation for the desired probability:

$$\mathbf{Pr}[\max_{i \in [N]} g_i \leqslant t_N] = \int_{-\infty}^{\infty} \psi(s) ds, \qquad (30)$$

$$\psi(s) \triangleq \phi(s) \Phi^N\left(\frac{t_N - \sqrt{\rho_0} s}{\sqrt{1 - \rho_0}}\right) ds, \qquad (31)$$

where $\phi(\cdot)$ and $\Phi(\cdot)$ are the standard Gaussian density and CDF, respectively. We will estimate this integral by splitting it into a couple pieces. To this end, we will bound the Gaussian CDF using Mills' inequality. For any $t > 0$, we have

$$\frac{t}{1 + t^2} \phi(t) \leqslant 1 - \Phi(t) = \Phi(-t) \leqslant \frac{1}{t} \phi(t).$$

We will need the following multiplicative estimate on $\Phi^N(\cdot)$, which holds for any $\delta \in [0, 1]$:

$$\Phi^N(\sqrt{2 \log N}(1 - \delta)) = (1 - \Phi(-\sqrt{2 \log N}(1 - \delta)))^N$$

$$\leqslant \left(1 - \frac{1}{\sqrt{2\pi} N^{(1-\delta)^2}(\sqrt{2 \log N}(1-\delta)+1)}\right)^N \qquad \text{(Mill's inequality)}$$

$$\leqslant \exp\left(-\frac{N^{1-(1-\delta)^2}}{\sqrt{2\pi}(\sqrt{2 \log N}+1)}\right). \qquad (32)$$

The above bound (32) yields nontrivial bounds only when $\delta > \frac{c \log \log N}{\log N}$ for a constant $c > \frac{1}{4}$, and decays superpolynomially if $c > \frac{3}{4}$.

These bounds motivate splitting (31) into a few different pieces, based on which term dominates the behavior of the integral $\psi$. To this end, let $c_N, d_N > 0$ be parameters we specify shortly. Then we split the integral into three pieces:

$$\int_{-\infty}^{-c_N} \psi(s) \, ds + \int_{-c_N}^{-d_N} \psi(s) \, ds + \int_{-d_N}^{\infty} \psi(s) \, ds \, .$$

As in Lopes & Yao (2022), we define $\alpha_0 \triangleq (\frac{1}{\rho_0} - 1)(1 - \delta_0)^2$ and $\beta_0 \triangleq \frac{\alpha_0}{1-\delta_0}$. Then the ultimate bound we want to prove is $\int_{-\infty}^{\infty} \psi(s) \, ds \leqslant O\left(N^{-\alpha_0}(\log N)^{\frac{\beta_0-1}{2}}\right)$. We explain the choice of $c_N, d_N > 0$ as follows. For succinctness, we introduce the following two functions on $\mathbb{R}$:

$$s_N(u) \triangleq -\sqrt{\frac{2(1-\rho_0) \log N}{\rho_0}} u = -\frac{\sqrt{2\alpha_0 \log N}}{1 - \delta_0} u$$

$$f_N(s) \triangleq \frac{t_N - \sqrt{\rho_0} s}{\sqrt{1 - \rho_0}},$$

and reparameterize the interval $[-c_N, -d_N]$ as another interval $\mathcal{I}$ on $u$ below.

1. We want $c_N$ to satisfy $\Phi(-c_N) = O\left(N^{-\alpha_0}(\log N)^{\frac{\beta_0-1}{2}}\right)$. Given this, the first term can be bounded by

$$\int_{-\infty}^{-c_N} \psi(s) \, ds \leqslant \int_{-\infty}^{-c_N} \phi(s) \, ds = \Phi(-c_N) \leqslant O\left(\frac{1}{N^{\alpha_0}}(\log N)^{\frac{\beta_0-1}{2}}\right).$$

By inverting Mills' inequality, we see that it suffices to pick

$$c_N \triangleq \sqrt{\frac{2(1-\rho_0) \log N}{\rho_0}}\left(1 - \delta_0 - \frac{\log \log N}{4 \log N}\right) \qquad (33)$$

$$= s_N\left(1 - \delta_0 - \frac{\log \log N}{4 \log N}\right).$$

Indeed, we have for sufficiently large $N$ that

$$\Phi(-c_N) \leqslant \frac{\phi(c_N)}{c_N} \leqslant \frac{\exp\left(-\alpha_0 \log N (1 - \frac{\log \log N}{4(1-\delta_0) \log N}^2)\right)}{\sqrt{2\pi}\sqrt{2(\frac{1}{\rho_0} - 1) \log N}(1 - o_N(1))}$$

$$\leqslant \sqrt{\frac{\rho_0}{1 - \rho_0}} N^{-\alpha_0}(\log N)^{\frac{\beta_0-1}{2}} \qquad (\beta_0 = \frac{\alpha_0}{1-\delta_0})$$

2. We pick $d_N$ such that $\Phi^N(f_N(-d_N)) \ll \frac{1}{N^{\alpha_0}}(\log N)^{\frac{\beta_0-1}{2}}$. Given this, by monotonicity we evidently have

$$\int_{-d_N}^{\infty} \psi(s) \, ds \leqslant \Phi^N(f_N(-d_N)) \ll \frac{1}{N^{\alpha_0}}(\log N)^{\frac{\beta_0-1}{2}}.$$

Owing to (32), to achieve the desired superpolynomial decay it suffices to pick $d_N \geqslant 0$ such that $\frac{t_N + \sqrt{\rho_0} d_N}{\sqrt{1 - \rho_0}} \leqslant \sqrt{2 \log N}(1 - (\frac{3}{4} + \varepsilon)\frac{\log \log N}{\log N})$, where $\varepsilon > 0$ is a constant.

Recalling that $t_N = \delta_0 \sqrt{2(1 - \rho_0) \log N}$, we see that $f_N(-d_N) = \sqrt{2 \log N}\delta_0 + \frac{\sqrt{\rho_0}}{\sqrt{1 - \rho_0}} d_N$. Hence the desired inequality holds by picking

$$d_N \triangleq \sqrt{\frac{2(1 - \rho_0) \log N}{\rho_0}}\left(1 - \delta_0 - (\tfrac{3}{4} + \varepsilon)\frac{\log \log N}{\log N}\right) \tag{34}$$

$$= s_N\left(1 - \delta_0 - (\tfrac{3}{4} + \varepsilon)\frac{\log \log N}{\log N}\right).$$

Given the above choices of $c_N, d_N$ in Eqs. (33) and (34), we see that the interval $\mathcal{I}$ that $u$ belongs to is $\mathcal{I} \triangleq [1 - \delta_0 - (\frac{3}{4} + \varepsilon)\frac{\log \log N}{\log N}, 1 - \delta_0 - \frac{1}{4} \cdot \frac{\log \log N}{\log N}]$. Hence, it is natural to reparameterize the integral in terms of $\eta \in [0, \frac{1}{2} + \varepsilon]$ via the following change of variables

$$u_N(\eta) \triangleq 1 - \delta_0 - \frac{(\eta + \frac{1}{4}) \log \log N}{\log N},$$

and an easy computation yields

$$\mathrm{d}s = s'_N(u_N(\eta))u'_N(\eta)\, \mathrm{d}\eta = \frac{\sqrt{2\alpha_0}}{1 - \delta_0} \cdot \frac{\log \log N}{\log N}\, \mathrm{d}\eta .$$

It is not hard to see that $u_N([0, \frac{1}{2} + \varepsilon]) = [-c_N, -d_N]$. Let us introduce the following abbreviations for the compositions of our changes of variable:

$$\widetilde{s}_N(\eta) \triangleq s_N(u_N(\eta))$$

$$\widetilde{f}_N(\eta) \triangleq f_N(\widetilde{s}_N(\eta)) = \sqrt{2 \log N}\left(1 - \frac{(\eta + \frac{1}{4}) \log \log N}{\log N}\right).$$

In this notation, we have

$$\int_{-c_N}^{-d_N} \psi(s)\, \mathrm{d}s = \frac{\sqrt{2\alpha_0}}{1 - \delta_0} \cdot \frac{\log \log N}{\log N} \int_0^{1/2+\varepsilon} \psi(\widetilde{s}_N(\eta))\, \mathrm{d}\eta .$$

Now the argument as in (Lopes & Yao, 2022, Eq. 4.22) shows that this integral is $O(N^{-\alpha_0}(\log N)^{\frac{\beta_0 - 1}{2}})$, which completes the proof of the first part of the theorem.[6] However, let us give an alternative proof which is a bit simpler. We will estimate the integral by performing a Riemann sum with subintervals of width $\tau > 0$, which we pick to satisfy $\beta_0 \tau < 1$ and such that $\tau$ evenly divides $\frac{1}{2} + \varepsilon$. Then, we have

$$\int_0^{1/2+\varepsilon} \psi(\widetilde{s}_N(\eta))\, \mathrm{d}\eta \leqslant \sum_{k=0}^{\frac{1/2+\varepsilon}{\tau}} \int_{k\tau}^{(k+1)\tau} \psi(\widetilde{s}_N(\eta))\, \mathrm{d}\eta .$$

By monotonicity, for any $k$ we have

$$\int_{k\tau}^{(k+1)\tau} \psi(\widetilde{s}_N(\eta))\, \mathrm{d}\eta \leqslant \tau \cdot \phi(\widetilde{s}_N((k+1)\tau))\Phi^N(\widetilde{f}_N(k\tau))$$

$$\leqslant C \cdot \tau \cdot N^{-\alpha_0}(\log N)^{\beta_0 \cdot (\frac{1}{2} + 2(k+1)\tau)} \exp\left(-(\log N)^{2k\tau}\right),$$

where $C$ is a universal constant. Here, the last line used $\phi(\widetilde{s}_N(\eta)) = O(N^{-\alpha_0}(\log N)^{2\beta_0 \cdot (\frac{1}{4} + \eta)})$ and $\Phi^N(\widetilde{f}_N(\eta)) = O(\exp(-\log N)^{2\eta})$.

If $2\beta_0(k+1)\tau < \frac{1}{2}$, then $\frac{\log \log N}{\log N}(\log N)^{2\beta_0(k+1)\tau} \ll (\log N)^{-\frac{1}{2}}$, as desired. On the other hand, if $2\beta_0(k+1)\tau \geqslant \frac{1}{2}$, then as $\beta_0 \tau < \frac{1}{8}$, we have $2k\tau \geqslant \frac{1}{4\beta_0} > 0$. Since $\exp\left(-(\log N)^{2k\tau}\right)$ dominates any polylog terms, so it is not hard to see that the total contribution here is $\ll N^{-\alpha_0}(\log N)^{\frac{\beta_0 - 1}{2}}$. Hence, we conclude that $\int_{-c_N}^{-d_N} \psi(s)\, \mathrm{d}s \leqslant \sqrt{\frac{\rho_0}{1 - \rho_0}} N^{-\alpha_0}(\log N)^{\frac{\beta_0 - 1}{2}}$, as desired. $\square$

---

[6]To be explicit, their calculation requires us to verify that $\sqrt{2 \log N}(\delta_0 + u_N(\eta)) = \omega(1)$, which is certainly true.

## F    EXPERIMENTS

In this section, we describe the simulations we conducted to validate the theory. We generated Gaussian data following the subset ensembles specified in the figures, and constructed two linear models from them: the MNI classifier and the simple averaging classifier. In the averaging classifier, we average over the positive examples of a label, which approximates the behavior of the first few iterations of gradient descent. In contrast, the MNI classifier governs the asymptotic behavior of gradient descent.

The weak-to-strong behavior for these two learning algorithms was compared to two other baselines: the weak accuracy for $f_{\text{weak}}$ and the accuracy for the strong model trained on $m$ clean labels: $f_{\text{strong}}$. The test accuracies were evaluated on $n_{\text{test}} = 100$ fresh datapoints. We ran $8$ independent trials to train $f_{\text{weak}}$ with $n = 50$ so that we could explore how the weak-to-strong behavior scales with $p$ and $u$. For each $f_{\text{weak}}$, we conducted 16 independent trials to train $f_{\text{w2s}}$.

We swept out $u$ using five equally spaced points in $[1, 1.3]$. In Figures 3 and 4, we show the results of the averaging and MNI experiments, respectively, for four different slices. In the top row, we show two slices where the theory predicts weak-to-strong generalization to occur for MNI, and in the bottom row we show two slices where the theory predicts failure of weak-to-strong generalization. The error bars show the estimated 95% CI over all sources of uncertainty in the inner and outer loop ($f_{\text{w2s}}$ and $f_{\text{weak}}$). In both the averaging and MNI plots, the theory successfully predicts whether weak-to-strong generalization occurs. Furthermore, in every plot the ground truth trained strong model $f_{\text{strong}}$ trained on $m$ clean labels has better test accuracy than $f_{\text{weak}}$ and $f_{\text{w2s}}$, as expected. Another interesting experimental observation is that the averaging classifier does significantly better than MNI in non-asymptotic settings. This corroborates the view of practitioners of the benefits of early-stopping for gradient descent.

## G    HEURISTIC CALCULATIONS

Recall from the definition of MNI that

$$f_{\text{strong}} = \boldsymbol{X}^\top \boldsymbol{A}^{-1} \boldsymbol{y},$$

where $\boldsymbol{y} \in \{\pm 1\}^n$ is a label vector generated by either the true feature $x_*$ or a weak feature $x_{\text{weak}}$ and $\boldsymbol{A} = \boldsymbol{X}\boldsymbol{X}^\top \in \mathbb{R}^{n \times n}$ is the Gram matrix. The key to our analysis is studying the survival and contamination of various features when the labels are possibly generated by another feature. Recall that we performed a basis change $\boldsymbol{X} \mapsto \boldsymbol{X}U$, so that the strong features are drawn iid from $N(0, \Lambda)$.

Writing the transformed data matrix now as

$$\boldsymbol{X} = \begin{bmatrix} \sqrt{\lambda_1}\boldsymbol{z}_1 & \sqrt{\lambda_2}\boldsymbol{z}_2 & \dots & \sqrt{\lambda_d}\boldsymbol{z}_d \end{bmatrix},$$

where each $\boldsymbol{z}_i \sim N(0, I_n)$, we obtain for any unit norm $v \in \mathbb{R}^D$ that

$$\mathsf{SU}(v) = \sum_{i \in [d]} \lambda_i \boldsymbol{z}_i^\top \boldsymbol{A}^{-1} \boldsymbol{y} \langle \boldsymbol{v}_i, v \rangle \tag{MNI}$$

$$\mathsf{CN}(v) = \sqrt{\sum_{i \in [d]} (\lambda_i \boldsymbol{z}_i^\top \boldsymbol{A}^{-1} \boldsymbol{y})^2 (1 - \langle \boldsymbol{v}_i, v \rangle^2)}. \tag{Orthonormality of $\boldsymbol{v}_i$}$$

Since $\boldsymbol{A}$ is close to $dI_d$ in the isotropic case, and we are working in the regime where PCA fails to extract the bi-level structure ($q + r > 1$), one could hope for the best and pretend that $\boldsymbol{A}^{-1} = \frac{1}{d}I_d$. This step is not rigorous, but we will justify these approximations in Appendix B.

Based on the decompositions in Definition 3, we study the case where $\boldsymbol{y} = \text{sgn}(\langle \boldsymbol{g}, \boldsymbol{w} \rangle)$ for some $\boldsymbol{w} \in R^D$ but we want to recover the planted direction $\boldsymbol{v} \in \mathbb{R}^D$. For a subspace $V \subseteq \mathbb{R}^D$ and a vector $\boldsymbol{u} \in \mathbb{R}^D$, let $\boldsymbol{u}_V$ denote the projection of $\boldsymbol{u}$ onto $V$. For axis-aligned subspaces $V \subseteq [d]$, this just corresponds to restricting to the coordinates in $V$. To simplify the heuristic calculation, we will make the following assumptions.

**Assumption 5.** *Let $S = [s]$ denote the spiked subspace after the basis change, and let $\alpha, \rho > 0$ be parameters possibly depending on $n$. For any vector $\boldsymbol{u} \in \mathbb{R}^d$, let $T_{\boldsymbol{u}} \triangleq \left\{ i \in S : |u_i| = \omega(\frac{1}{\sqrt{n}}) \right\}$*

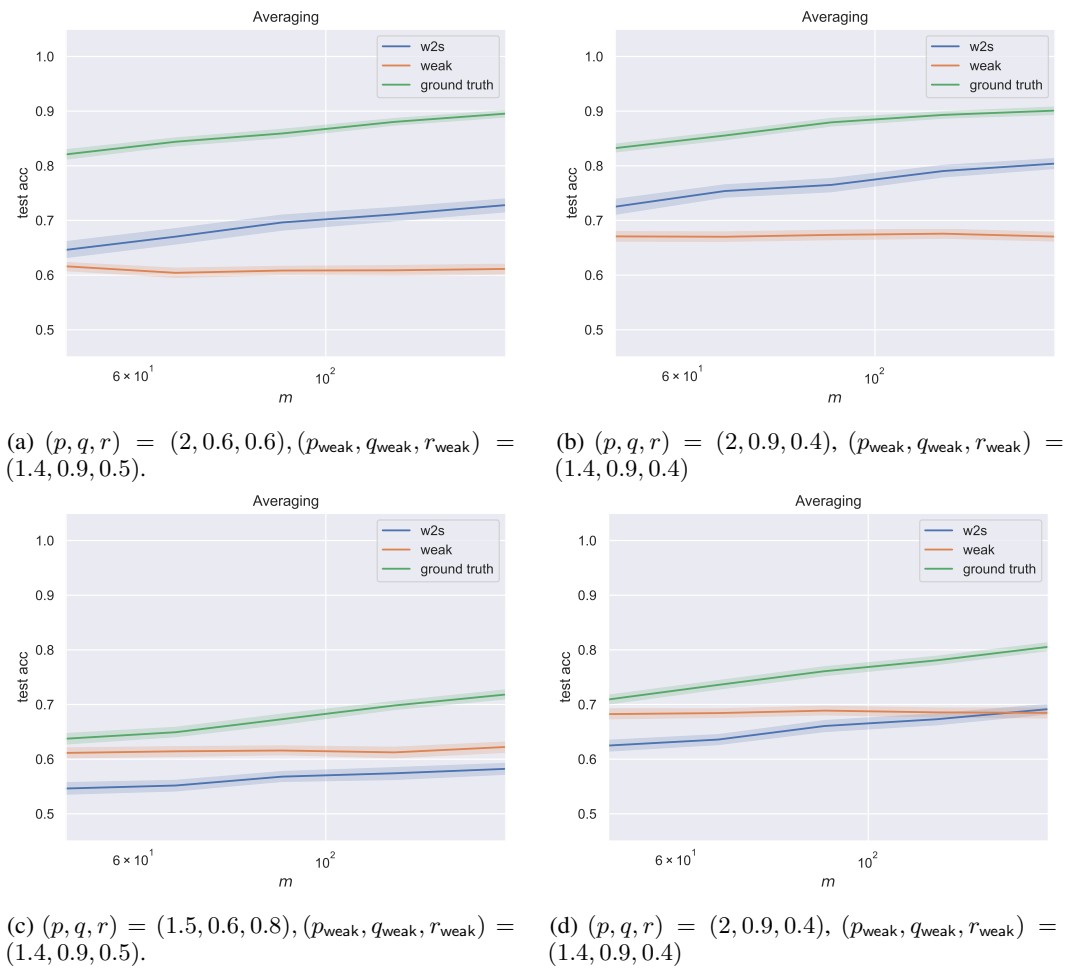

(a) $(p, q, r) = (2, 0.6, 0.6), (p_{\text{weak}}, q_{\text{weak}}, r_{\text{weak}}) = (1.4, 0.9, 0.5)$.

(b) $(p, q, r) = (2, 0.9, 0.4), (p_{\text{weak}}, q_{\text{weak}}, r_{\text{weak}}) = (1.4, 0.9, 0.4)$

(c) $(p, q, r) = (1.5, 0.6, 0.8), (p_{\text{weak}}, q_{\text{weak}}, r_{\text{weak}}) = (1.4, 0.9, 0.5)$.

(d) $(p, q, r) = (2, 0.9, 0.4), (p_{\text{weak}}, q_{\text{weak}}, r_{\text{weak}}) = (1.4, 0.9, 0.4)$

Figure 3: Comparison of test accuracies for four different models using averaging training. The $x$-axis plots $m$, the number of additional labeled datapoints. The models are trained using class averaging, which approximates the behavior of the initial few gradient descent iterations. Note how the weak model has low accuracy, whereas the weak-to-strong model and ground truth have higher accuracies that increase as $m$ increases. The top row Figures 3a and 3b are in a regime where we predict MNI weak-to-strong generalization to succeed, whereas the bottom row Figures 3c and 3d depict regimes where we expect MNI weak-to-strong generalization to fail.

*denote the spiked coordinates where $\mathbf{u}$ is large, and $R_{\mathbf{u}} = S \setminus T_{\mathbf{u}}$ denote the spiked coordinates where $\mathbf{u}$ is small.*

*We assume the following holds for $T = T_{\mathbf{w}}$ and $R = R_{\mathbf{w}}$:*

(1) *We have $\langle \mathbf{v}_T, \mathbf{w}_T \rangle = \alpha \|\mathbf{v}_T\|_2 \|\mathbf{w}_T\|_2$. In other words, $\mathbf{v}$ and $\mathbf{w}$ have correlation $\alpha$ restricted to the heavy coordinates for $\mathbf{w}$.*

(2) *We have $\sum_{i \in T} v_i^2 w_i^2 = \rho^2 \|\mathbf{w}_T\|_2^2$. Note that by $L^p$ norm inequalities we always have $\rho^2 \leqslant 1$.*

(3) *We have $|R| = \Omega(s)$, i.e. a constant fraction of $\mathbf{w}$'s spiked coordinates are small.*

**Remark G.1.** *When $\mathbf{v} = \mathbf{w}$, Item (2) can be thought of as a relaxed notion of $\mathbf{v}$ being 1-sparse; for a given $\rho$ one should roughly think of $\mathbf{v}$ as being $\frac{1}{\rho}$-sparse.*

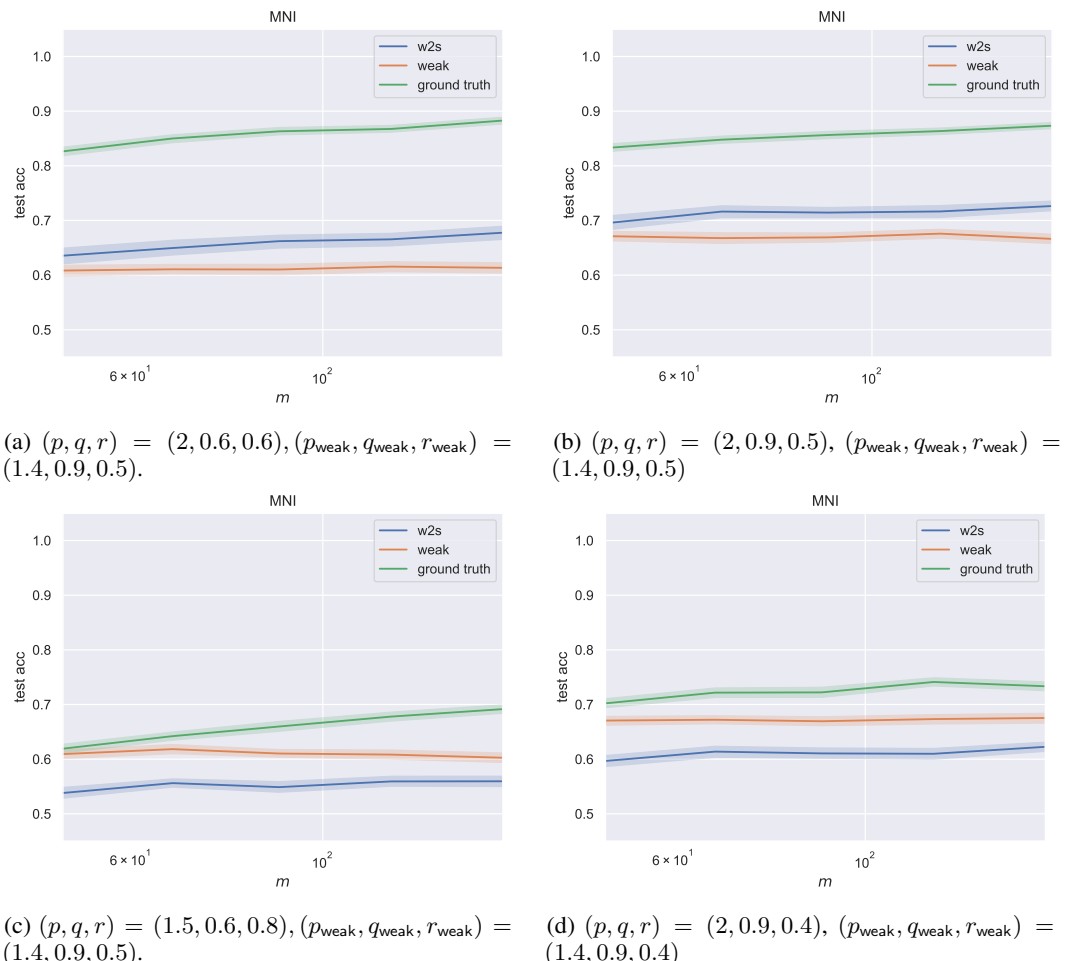

(a) $(p, q, r) = (2, 0.6, 0.6)$, $(p_{\text{weak}}, q_{\text{weak}}, r_{\text{weak}}) = (1.4, 0.9, 0.5)$.

(b) $(p, q, r) = (2, 0.9, 0.5)$, $(p_{\text{weak}}, q_{\text{weak}}, r_{\text{weak}}) = (1.4, 0.9, 0.5)$

(c) $(p, q, r) = (1.5, 0.6, 0.8)$, $(p_{\text{weak}}, q_{\text{weak}}, r_{\text{weak}}) = (1.4, 0.9, 0.5)$.

(d) $(p, q, r) = (2, 0.9, 0.4)$, $(p_{\text{weak}}, q_{\text{weak}}, r_{\text{weak}}) = (1.4, 0.9, 0.4)$

Figure 4: Comparison of MNI test accuracies for four different models. Observe how the weak-to-strong accuracy increases as $m$ increases. Again, the top row Figures 4a and 4b are in a regime where we predict MNI weak-to-strong generalization to succeed, whereas the bottom row Figures 4c and 4d depict regimes where we expect MNI weak-to-strong generalization to fail. The plots corroborate these theoretical predictions.

**Survival bound.** Recall that $\boldsymbol{y} = \text{sgn}(\langle \boldsymbol{g}, \boldsymbol{w} \rangle)$. By applying the noise stability formula again and the fact that $z_{ij} \sim N(0, 1)$, we deduce that

$$
\begin{aligned}
\mathbf{E}[\boldsymbol{z}_i^\top \boldsymbol{y}] &= n\mathbf{E}[z_{ij} y_j] \\
&= \mathbf{Pr}[\text{sgn}(z_{ij}) = y_j]\mathbf{E}[|z_{ij}|] - (1 - \mathbf{Pr}[\text{sgn}(z_{ij}) = \boldsymbol{y}_j])\mathbf{E}[|z_{ij}|] \\
&= \sqrt{\tfrac{2}{\pi}}(2\mathbf{Pr}[\text{sgn}(z_{ij}) = y_j] - 1) \\
&= (\tfrac{2}{\pi})^{3/2} \arcsin w_i,
\end{aligned}
$$

where in the second to last line we have used the fact that the expected magnitude of a standard Gaussian is $\sqrt{2/\pi}$, and in the last line we have used the noise stability formula. By standard concentration inequalities, the deviations will be of order $O(\sqrt{n})$. As the behavior of $\frac{2}{\pi} \arcsin(x) \approx \frac{2}{\pi}x$ for small $x$, we deduce that the expectation will dominate whenever $|w_i| \gg \frac{1}{\sqrt{n}}$).

We will now plug in the bi-level scaling. Recall that $\lambda_i = \lambda_F = \frac{ad}{s}$ for $i \in [s]$ and $\lambda_i = \lambda_U = \frac{(1-a)d}{d-s}$ for $i > s$.

Thus, ignoring constants, with high probability we should have

$$\mathsf{SU}(\boldsymbol{v}|\boldsymbol{w}) = \frac{\lambda_F n}{d} \sum_{i \in T_{\boldsymbol{w}}} v_i \arcsin w_i \pm \frac{\lambda_F \sqrt{n}}{d} \sum_{i \in R_{\boldsymbol{w}}} v_i \pm \frac{\lambda_U \sqrt{n}}{d} \sum_{i > s} v_i .$$

Observe that

$$\frac{2}{\pi}|x| \leqslant \frac{2}{\pi}|\arcsin x| \leqslant |x|,$$

for all $x \in [-1, 1]$.

Consequently, for $i \in T_{\boldsymbol{w}}$, each summand contributes $v_i \arcsin(w_i) \approx \Theta(v_i w_i)$.[7] Thus, by Item (1) and the definition of $\lambda_F$, the first term is $\Theta(\alpha \cdot \frac{an}{s} \cdot \|\boldsymbol{v}_{T_{\boldsymbol{w}}}\|_2 \|\boldsymbol{w}_{T_{\boldsymbol{w}}}\|_2)$.

For the second term, we will upper bound its magnitude by

$$\frac{\lambda_F \sqrt{n}}{d} \left| \sum_{i \in R_{\boldsymbol{w}}} v_i \right| \leqslant \frac{an}{s} \cdot \frac{\|\boldsymbol{v}_{R_{\boldsymbol{w}}}\|_1}{\sqrt{n}}$$

$$\leqslant \frac{an}{s} \cdot \sqrt{\frac{s}{n}} \cdot \|\boldsymbol{v}_{R_{\boldsymbol{w}}}\|_2 \qquad\qquad (R_{\boldsymbol{w}} \subseteq S)$$

$$\leqslant \frac{an}{s} \cdot \sqrt{\frac{s}{n}}. \qquad\qquad (\|\boldsymbol{v}\|_2 = 1)$$

Finally, for the third term, since $\left|\sum_{i>s} v_i\right| \leqslant \|\boldsymbol{v}\|_1 \leqslant \sqrt{d}$, after plugging in the definition of $\lambda_U$, we obtain the asymptotics

$$\mathsf{SU}(\boldsymbol{v}|\boldsymbol{w}) \asymp \frac{an}{s} \cdot \left( \alpha \cdot \|\boldsymbol{v}_{T_{\boldsymbol{w}}}\|_2 \|\boldsymbol{w}_{T_{\boldsymbol{w}}}\|_2 \pm \sqrt{\frac{s}{n}} \right) \pm \sqrt{\frac{n}{d}}. \qquad (35)$$

Note that for the first term to dominate, we must have $\alpha = \Omega(\frac{s}{n})$. Also, if we want to improve our estimate on the second term, we can further split it by $T_{\boldsymbol{v}}$, we can gain and get deviations of order

$$\left| \sum_{i \in R_{\boldsymbol{w}} \cap T_{\boldsymbol{v}}} v_i + \sum_{i \in R_{\boldsymbol{w}} \cap R_{\boldsymbol{v}}} v_i \right| \leqslant \|\boldsymbol{v}_{R_{\boldsymbol{w}} \cap T_{\boldsymbol{v}}}\|_1 + \frac{|R_{\boldsymbol{v}}|}{\sqrt{n}} \qquad (\text{Definition of } R_{\boldsymbol{v}})$$

$$\leqslant \|\boldsymbol{v}_{R_{\boldsymbol{w}} \cap T_{\boldsymbol{v}}}\|_1 + \frac{s}{\sqrt{n}},$$

yielding an ultimate relative deviation of $\frac{\|\boldsymbol{v}_{R_{\boldsymbol{w}} \cap T_{\boldsymbol{v}}}\|_1}{\sqrt{n}} + \frac{s}{n}$. This is significantly better if, say $|T_{\boldsymbol{v}}| = o(s)$, as it allows us to beat the contamination bounds which have relative deviations $\sqrt{\frac{s}{n}}$.

**Contamination bound.** For succinctness, introduce the shorthand $h_i^2 = (1 - v_i^2) \langle \boldsymbol{z}_i, \boldsymbol{y} \rangle^2$. Then the squared contamination is

$$\mathsf{CN}(\boldsymbol{v}|\boldsymbol{w})^2 = \sum_{i \in [d]} \lambda_i^2 (1 - v_i^2) \cdot \boldsymbol{y}^\top \boldsymbol{A}^{-1} \boldsymbol{z}_i \boldsymbol{z}_i^\top \boldsymbol{A}^{-1} \boldsymbol{y}$$

$$\approx \frac{1}{d^2} \sum_{i \in [d]} \lambda_i^2 h_i^2 \qquad\qquad (\boldsymbol{A}^{-1} \approx \frac{1}{d} I_d)$$

$$= \frac{\lambda_F^2}{d^2} \sum_{i \in T_{\boldsymbol{w}}} h_i^2 + \frac{\lambda_F^2}{d^2} \sum_{i \in R_{\boldsymbol{w}}} h_i^2 + \frac{\lambda_U^2}{d^2} \sum_{i > s} h_i^2$$

$$\leqslant \frac{\lambda_F^2 n^2}{d^2} \sum_{i \in T_{\boldsymbol{w}}} (1 - v_i^2)(\tfrac{2}{\pi} \arcsin w_i)^2 + \frac{\lambda_F^2 n}{d^2} \sum_{i \in R_{\boldsymbol{w}}} (1 - v_i^2) + \frac{\lambda_U^2 n}{d^2}(d - s),$$

where in the last line we have used the observation that the expectation of $h_i$ dominates if and only if $i \in T_{\boldsymbol{w}}$.

---

[7]The reason this is not an equality is that there might be some heavy $w_i's$ which disagree in sign with $v_i's$, but for the settings we consider this estimate will be true.

The first term can be bounded (up to constants) as

$$\left(\frac{an}{s}\right)^2 \sum_{i \in T_{\boldsymbol{w}}} (1 - v_i^2) w_i^2 = \left(\frac{an}{s}\right)^2 \|\boldsymbol{w}_{T_{\boldsymbol{w}}}\|_2^2 (1 - \rho^2). \qquad \text{(Item (2))}$$

For the second term, we can bound up to constants as

$$\left(\frac{an}{s}\right)^2 \cdot \frac{1}{n} \cdot \sum_{i \in R_{\boldsymbol{w}}} (1 - v_i^2) = \left(\frac{an}{s}\right)^2 \cdot \frac{|R_{\boldsymbol{w}}| - \|\boldsymbol{v}_{R_{\boldsymbol{w}}}\|_2^2}{n}$$

$$\leqslant \left(\frac{an}{s}\right)^2 \cdot \frac{|R_{\boldsymbol{w}}|}{n}. \qquad (\|\boldsymbol{v}\|_2 = 1)$$

Finally, the third term can be bounded by $\frac{n}{d}$. Putting these together, we conclude that

$$\mathsf{CN}(\boldsymbol{v}|\boldsymbol{w}) \asymp \left(\frac{an}{s}\right)\left(\sqrt{1-\rho^2}\|\boldsymbol{w}_{T_{\boldsymbol{w}}}\|_2 + \sqrt{\frac{|R_{\boldsymbol{w}}|}{n}}\right) + \sqrt{\frac{n}{d}}. \qquad (36)$$

Let $\mu_n = \frac{an}{s}$. In our regime, $\mu_n \ll 1$ because $q + r > 1$. Combining Equations (35) and (36) yields

$$\frac{\mathsf{SU}(\boldsymbol{v}|\boldsymbol{w})}{\mathsf{CN}(\boldsymbol{v}|\boldsymbol{w})} \asymp \frac{\mu_n \cdot \left(\alpha\|\boldsymbol{v}_{T_{\boldsymbol{w}}}\|_2\|\boldsymbol{w}_{T_{\boldsymbol{w}}}\|_2 \pm \frac{\|\boldsymbol{v}_{R_{\boldsymbol{w}} \cap T_{\boldsymbol{v}}}\|_1}{\sqrt{n}} \pm \frac{s}{n}\right) \pm \sqrt{\frac{n}{d}}}{\mu_n \left(\sqrt{1-\rho^2}\|\boldsymbol{w}_{T_{\boldsymbol{w}}}\|_2 + \sqrt{\frac{|R_{\boldsymbol{w}}|}{n}}\right) + \sqrt{\frac{n}{d}}} \qquad (37)$$

Hence, for the survival to contamination ratio to grow with $n$, we need

$$\alpha\|\boldsymbol{v}_{T_{\boldsymbol{w}}}\|_2 \gg \sqrt{1-\rho^2} \qquad \text{(Weak supervision)}$$

$$\alpha\|\boldsymbol{v}_{T_{\boldsymbol{w}}}\|_2\|\boldsymbol{w}_{T_{\boldsymbol{w}}}\|_2 \gg \sqrt{\frac{|R_{\boldsymbol{w}}|}{n}} \qquad \text{(Favored contamination)}$$

$$\mu_n\alpha\|\boldsymbol{v}_{T_{\boldsymbol{w}}}\|_2\|\boldsymbol{w}_{T_{\boldsymbol{w}}}\|_2 \gg \sqrt{\frac{n}{d}}$$

Let us put these scalings together to predict the scaling regimes for weak-to-strong generalization. For strong generalization, we have $\boldsymbol{v} = \boldsymbol{v}_*$ and $\boldsymbol{y} = \mathrm{sgn}(\langle \boldsymbol{g}, \boldsymbol{v}_* \rangle)$. From the discussion in Section 3.1, we know that the strong learner generalizes if

$$\frac{\mathsf{SU}(\boldsymbol{v}_*|\boldsymbol{v}_*)}{\mathsf{CN}(\boldsymbol{v}_*|\boldsymbol{v}_*)} = \omega_n(1),$$

and fails to generalize if the ratio is $o_n(1)$. Under Assumption 1, we have $\boldsymbol{v}_* = e_1$, so $T_{\boldsymbol{v}} = T_{\boldsymbol{w}} = \{1\}$, $\alpha = 1$, and $\rho = 1$, and the expression simplifies to

$$\frac{\mathsf{SU}(\boldsymbol{v}_*|\boldsymbol{v}_*)}{\mathsf{CN}(\boldsymbol{v}_*|\boldsymbol{v}_*)} \asymp \frac{\mu_n}{\mu_n \cdot \sqrt{\frac{s}{n}} + \sqrt{\frac{n}{d}}},$$

which under the bi-level parameter scaling verifies the conditions for ground truth supervision.

This completes the proof sketch; it remains to justify the above estimates rigorously. In the subsequent subsections, we will assume that the above scalings of the survival and contamination are correct and use them to deduce that the 1-sparse assumption is necessary to get a sharp transition in the test error. These calculations can be upgraded to rigorous proofs using the tools are developed in Appendix B.

### G.1 THE NECESSITY OF 1-SPARSE ASSUMPTION

Let's suppose we get clean labels from $\mathrm{sgn}(\langle \boldsymbol{g}, \boldsymbol{v} \rangle)$ and want to learn the unit vector $\boldsymbol{v}$. We will abbreviate $T = T_{\boldsymbol{v}} = T_{\boldsymbol{w}}$. In this case, we have $\alpha = 1$ and $\rho^2 = \frac{\|\boldsymbol{v}_T\|_4^4}{\|\boldsymbol{v}_T\|_2^2}$ in Assumption 5.

**Lemma G.2.** *Suppose we are given labels according to $\boldsymbol{v}$ and want to learn $\boldsymbol{v}$. Then, the survival to contamination ratio is $\omega_n(1)$ only if*

$$\frac{\|\boldsymbol{v}_T\|_4^4}{\|\boldsymbol{v}_T\|_2^2} = 1 - o(1).$$

*In particular, the above condition holds only if the following two conditions hold:*

(1) $\|\boldsymbol{v}_T\|_2^2 = 1 - o(1)$.

(2) *For each $i \in T$, either $v_i = o(1)$ or $v_i = 1 - o(1)$.*

*The upshot is that having 1-sparse labels is necessary for obtaining asymptotically perfect generalization.*

*Proof.* Hence, focusing only on the survival terms coming from $T$, which are the only relevant coordinates for learning, and lower bounding the contamination with just the $T$ terms, we have

$$\frac{\mathsf{SU}(\boldsymbol{v}|\boldsymbol{v})}{\mathsf{CN}(\boldsymbol{v}|\boldsymbol{v})} \leqslant \frac{\frac{an}{s}\|\boldsymbol{v}_T\|_2^2}{\frac{an}{s}\sqrt{\|\boldsymbol{v}_T\|_2^2 - \|\boldsymbol{v}_T\|_4^4}} = \frac{\|\boldsymbol{v}_T\|_2}{\sqrt{1 - \frac{\|\boldsymbol{v}_T\|_4^4}{\|\boldsymbol{v}_T\|_2^2}}} \leqslant \frac{1}{\sqrt{1 - \frac{\|\boldsymbol{v}_T\|_4^4}{\|\boldsymbol{v}_T\|_2^2}}},$$

where the last inequality used the fact that $\boldsymbol{v}$ is unit norm. This proves the first necessary condition.

We show that if $\|\boldsymbol{v}_T\|_4^4/\|\boldsymbol{v}_T\|_2^2 = 1 - o(1)$, then the second set of necessary conditions hold.

Indeed, for the first claim, suppose $\|\boldsymbol{v}_T\|_2^2 \leqslant 1 - \varepsilon$ for some constant $\varepsilon > 0$. The $L^p$ norm inequalities imply that $\|\boldsymbol{v}_T\|_4^4 \leqslant \|\boldsymbol{v}_T\|_2^4 \leqslant (1 - \varepsilon)\|\boldsymbol{v}_T\|_2^2$, so the ratio is at most $1 - \varepsilon$, a contradiction. For the second claim, suppose instead there is a coordinate $i$ with $v_i^2 = 1 - \varepsilon$ for some constant $\varepsilon \in (0, 1)$. Then, we have $\left\|\boldsymbol{v}_{T\setminus\{i\}}\right\|_4^4 \leqslant \left\|\boldsymbol{v}_{T\setminus\{i\}}\right\|_2^4 \leqslant \varepsilon$, but then $\|\boldsymbol{v}_T\|_4^4 \leqslant (1 - \varepsilon)^2 + \varepsilon \leqslant 1 - \Omega(\varepsilon)$, a contradiction.

$\square$

