# OpenReview forum: "Provable weak-to-strong generalization via benign overfitting"
_ICLR.cc/2025/Conference — ICLR 2025 Poster_

### Official Review · Reviewer_ww6B · 2024-10-31

**Soundness:** 3
**Presentation:** 2
**Contribution:** 3
**Rating:** 6
**Confidence:** 2

**Summary:**

This paper studies a toy-model for weak-to-strong generalization. They show that under the assumptions of their toy-model, two asymptotic phases occur for the student: (1) it is able to successfully generalize or (2) the student resorts to effectively random guessing. The authors also try to extend their results to weak-to-strong multiclass classification and derive new lower tail inequalities for the max of correlated gaussians.

**Strengths:**

- The paper studies a phenomenon that has been empirically observed and thus relevant to practice
- The results and proof techniques seem non-trivial and interesting

**Weaknesses:**

I find the organization and presentation a bit confusing and hard to parse. In particular, Theorem 3.3 is hard to interpret without referencing the Desiradatum outlined in Section 2. Ideally, Theorem 3.3 should be standalone and at the very least, the variables in Theorem 3.3 like $\tau_{weak}, p_{weak}, ...$ should be defined. In addition, and in my opinion, the notation and current presentation of the result doesn't really make it seem like this is a "simple, toy model", given how many free variables there are to keep track of.  One possible fix is to give more intuition and less notation about the toy-model in the main text, and push the details into the Appendix. For example, I think it would be really helpful to have an informal, non-rigorous theorem summarizing the main result in the Main Contributions section.

In addition, I am not sure what to take away from this paper. It is nice that you found a toy example, where you can provide rigorous evidence of the empirical phenomena of weak-to-strong generalization. However, I am not convinced this toy model is realistic/relevant to practice, even after reading the Modeling assumptions in the Discussion. In short, it would be nice if the authors can answer:
-  **why** one should care about finding a "simple, concrete theoretical setting where we can provably exhibit different phases of weak-to-strong generalization?"
- what can I take away from this result?

**Questions:**

See weaknesses above. It would be nice if the authors can address these.

---

> ### Author Response · Authors · 2024-11-18
> **Rebuttal**
>
> We thank the reviewer for their feedback. We address their questions below.
> > Ideally, Theorem 3.3 should be standalone and at the very least, the variables in Theorem 3.3 like $\tau_{\mathsf{weak}}$, $p_{\mathsf{weak}}$ should be defined.
> - We agree that it is better to have theorem statements be more self-contained. Stating the theorem precisely requires using a decent amount of notation, and to reduce redundancy we have set up the data assumptions and notations explicitly in Theorem 3.2. For example, we have explicitly defined $\tau_{\mathsf{weak}}$ in line 419.
> - In the revision, we have amended the tags and wording in Theorem 3.3, especially around the two conditions in lines 421-423. It should hopefully be more intuitive what each condition is referring to. Also, we have reinforced the definitions of $\tau_{\mathsf{strong}}$ and $\tau_{\mathsf{weak}}$.
>
> > The notation and current presentation of the result doesn't really make it seem like this is a "simple, toy model", given how many free variables there are to keep track of. One possible fix is to give more intuition and less notation about the toy-model in the main text, and push the details into the Appendix.
> - We agree that there are many parameters floating around. However, we also believe that this bi-level covariance is essentially the simplest parameterization of low-rank data that can be tractably studied from a theoretical perspective.
> - In particular, these parameters can be interpreted as follows: $p$ specifies the ambient dimension $d$, $r$ specifies the dimension $s$ of the low-rank subspace, $q$ specifies the strength $a$ of the spike. A full specification of a low-rank data distribution needs to specify these three parameters.
> - We will try to include some more intuition about the main result in the revision, but we do think it is still important to have a fully precise statement in the main text. That way, a reader can know what is the exact statement being claimed without chance of ambiguity.
>
> >  For example, I think it would be really helpful to have an informal, non-rigorous theorem summarizing the main result in the Main Contributions section.
> - In the main contributions, we have emphasized the informal statement of the main result.
>
> > why one should care about finding a "simple, concrete theoretical setting where we can provably exhibit different phases of weak-to-strong generalization?"
>
> - Weak-to-strong generalization is an interesting empirical phenomenon in ML models which might be important for scaling up post-training for foundation models. Human-labeled data is expensive to procure, but there is also no guarantee that using weakly-labeled data would scale well. Our theoretical results give provable guarantees for these kinds of training schemes in a toy setting. Moreover, because the setting is simple and theoretical, it is easier to trace down and explain exactly why the training strategy works.
>
> > what can I take away from this result?
>
> - One intuitive takeaway is that weak-to-strong generalization occurs based on how the strong model implicitly “knows” that certain latent directions in the weak model’s activations are more important than others. The weak model makes errors because its representations are low quality, but the key lies in the way its representations relate to the strong model's representations. Presumably, the reason the strong model's representations are better has to do with the way that it was pretrained as well as its own architecture.
>
> - Another takeaway regards our insights about multilabel vs multiclass training. In particular, we show that using multilabel weak supervision can do better than multiclass clean supervision! This suggests that in order to get effective weak supervision, whenever possible, one should use the logits / soft labels from the weak model to supervise the strong model. Because we can establish this rigorously in the classification-style setting here, it suggests that we should look for how to do a counterpart of this in the LLM setting (or more generically for generative models). Having access to a weak model for providing supervision opens up more possibilities than present with human-labeled data.

---

> > ### Comment · Reviewer_ww6B · 2024-11-20
> >
> > I thank the authors for their response. I have increased my score to a 6.

---

### Official Review · Reviewer_P4ZD · 2024-11-02

**Soundness:** 3
**Presentation:** 2
**Contribution:** 3
**Rating:** 8
**Confidence:** 2

**Summary:**

In this work, the authors provide theretical justification for the empirically observed phenomenon of weak to strong generalization. In this setting, a weak learner is used to created labelled examples (from unlabelled training data) that is used to further train a stronger model. The intuition is that the weak learner has learnt some useful information about the ground truth and hence the pseudolabels it generates will actually enable generalization. The authors prove that this weak to strong generalization has two phases: (1) when the number of pseudolabelled examples is less than some threshold, the strong learner behaves like a random guesser, (2) beyond the threshold the strong learner achieves perfect generalization. A technically interesting tool that they use is a new lower tail for the max of correlated gaussians which could be of independent interest.

**Strengths:**

1) This work addresses the important problem of obtaining theoretical justification for a frequently encountered empirical phenomenon
2) The lower tail for max of correlated gaussians is an interesting result.

**Weaknesses:**

See questions.

**Questions:**

21) What the the word "represent" mean in Desiredata 1.(ii).
2) What is the significance of the bi-level-ensemble?
3) What is $t$ in Theorem 3.1?
4) Is there a reason for choosing a halfspace for the ground truth? Does this analysis extend to other concepts. Is there a similar notion for regression (rather than classification)?

---

> ### Author Response · Authors · 2024-11-18
> **Rebuttal**
>
> We thank the reviewer for their positive review. Below, we address their questions.
>
> > What the the word "represent" mean in Desiredata 1.(ii).
> - Here we meant that the strong model can perfectly simulate the weak model. In other words, the strong model’s capabilities are a superset of those of the weak model. For the linear setting, this just means that the weak features are in the span of the strong features.
>
> > What is the significance of the bi-level-ensemble?
> - The bi-level ensemble is a minimal instantiation of a spiked covariance model that fully specifies all the aspects of the covariance of the data, i.e. a toy model for low-rank structure. You can also think of it as a parameterized linear manifold hypothesis. The bi-level ensemble has been studied in previous works as a way to state cleaner theoretical results for benign overfitting [1-4].
>
> > What is $t$ in Theorem 3.1?
> - As mentioned in the response to Reviewer Ns5A, $t \in [0, s)$ controls the number of label classes in the multiclass problem: $k = n^t$, following Definition 2. We have updated the wording of the theorem to make this more explicit.
>
> > Is there a reason for choosing a halfspace for the ground truth? Does this analysis extend to other concepts. Is there a similar notion for regression (rather than classification)?
> - (Shifted) halfspaces are fundamental in the study of classification. For example, even complicated neural network architectures for multiclass / binary classification are predicated on halfspaces, since the last layer is usually a linear head.
> The analysis would not directly work beyond the linear setting, as we use the explicit analytic form of the minimum $\ell_2$-norm interpolator.
> - For regression, the same linear model that gives a halfspace can also be used as a regression model, if one doesn’t apply softmax. Our analysis should go through in this case: previous work [3] already developed the tools to tightly characterize the regression setting.
>
> [1] Wang, K., & Thrampoulidis, C. (2022). Binary classification of gaussian mixtures: Abundance of support vectors, benign overfitting, and regularization. SIAM Journal on Mathematics of Data Science, 4(1), 260-284.
>
> [2] Wang, K., Muthukumar, V., & Thrampoulidis, C. (2021). Benign overfitting in multiclass classification: All roads lead to interpolation. Advances in Neural Information Processing Systems, 34, 24164-24179.
>
> [3] Muthukumar, V., Narang, A., Subramanian, V., Belkin, M., Hsu, D., & Sahai, A. (2021). Classification vs regression in overparameterized regimes: Does the loss function matter?. Journal of Machine Learning Research, 22(222), 1-69.
>
> [4] Wu, D., & Sahai, A. (2024). Precise asymptotic generalization for multiclass classification with overparameterized linear models. Advances in Neural Information Processing Systems, 36.

---

> > ### Comment · Reviewer_P4ZD · 2024-11-21
> > **Reply to rebuttal**
> >
> > I thank the authors for their response. I have left my score unchanged

---

### Official Review · Reviewer_vRgg · 2024-11-03

**Soundness:** 3
**Presentation:** 2
**Contribution:** 3
**Rating:** 6
**Confidence:** 3

**Summary:**

The papers identifies a specific setting under which weak to strong generalization occurs. Consider a strong model that learns a classifier on strong features of the data by supervised learning on $m$ weak labels given by a weak model that was trained on weak features on $n$ clean labels. Then weak to strong generalization implies that

Condition 1):  The strong model has perfect classification accuracy whereas the weak model has close to random accuracy.

Condition 2): The generalization is due to weak labels, i.e. if the strong model was only trained on $n$ clean labels, there is no generalization.

The setting is as follows: A learner observes features distributed according to a Gaussian distribution, $x \sim N(0, \Lambda)$ where $\Lambda$ is diagonal covariance matrix following a bilevel ensemble parameterization
\begin{equation}\lambda_j = \lambda_F =  \frac{ad}{s} \text{ for } 1 \leq j \leq s \text{ otherwise } \lambda_j = \lambda_U = \frac{(1-a)d}{d-s}\end{equation}
where $d = n^p, s= n^r, a = n^{-q}$ and $p > 1; q, r >0; q+ r < p$. For multiclass setting, classes are further scaled as $k = c_k n^t$ for some $t<r$.  The strong model observes features given by some $p, q, r$ and weak model observes features  characterized through $p_{weak}, q_{weak}, r_{weak}$. In particular the strong features $x_{strong}$ and weak features $x_{weak}$ are given as
$$ x_{strong} = N(0, \lambda_F I_{[s]} + \Lambda_U I_{[d]/[s]}) $$
$$ x_{weak} = N(0, \lambda_{F, weak} \Pi_S + \Lambda_{U, weak} \Pi_T)$$
for some subsets $S \subseteq [s], T \subseteq [d]/[s]$ and $\Pi_S$ denotes projection onto axis aligned subspace indexed by $S$. $\lambda_{F, weak}  = \frac{a_{weak}d_{weak}}{s_{weak}}$ and $\Lambda_{U, weak} =  \frac{(1-a_{weak})d_{weak}}{d_{weak}-s_{weak}}$.

 The true labels are given by $y = \text{sign}(x_1)$ for binary classification and $y = \arg\max_k (x_1, \dots x_K)$ for $K$ way classification.



In this parameterized setting, the authors show that there is a particular regime of number of weak labels $m$ provided by the weak model (for certain regimes of $p, q, r, p_{weak}, q_{weak}, r_{weak}$) where weak to strong generalization occurs (condition 1) holds). The conditions (for binary classification) are given by (assuming $m = n^u$)

1. $u + \min(1 -r,  p + 1 - 2(q + r)) > q_{weak}+r_{weak} > (p_{weak} + 1)/ 2$
2. $p + 1   >  (q + r + q_{weak} + r_{weak})$
3. $u < (p + 1 + q + r  - (q_{weak} + r_{weak})/ 2)$

Further the classification error of strong learner trained on $n$ cleaned labels is shown to be depend as
$$1/2 - 1/\pi \arctan (\Theta(n^{p+1 - 2(q+r)}))$$

Thus they claim one can identify regimes under which condition 2) also holds (possibly when $p+1 - 2(q+r) << 1$) although no details are provided).

Further they provide an informal claim and details in appendix that there exists some regime for multi class setting.

**Strengths:**

Exact characterization of the regime where weak to strong generalization occurs in terms of parameters of the covariance matrix of strong and weak features.

**Weaknesses:**

Most of the important details are pushed into appendix. The main body only contains one useful theorem which identifies a certain condition where condition 1) of weak to strong generalization holds. Setting for condition 2) and multi class settings are merely mentioned as claims. The main body also does not provide proof sketch or provide insights into the proof of the theorem.

**Questions:**

Suggestions:

1. Reduce the introduction - it currently spans 2 pages.
2. Figure 1 is useless.
3. The section on data model was not particularly needed. Page 5 and 6 can be compressed into 1 or 2 paragraphs.
4. Include some experiments in main body.

In general the paper is quite verbose, it can be compressed substantially and content moved back into main body.

---

> ### Author Response · Authors · 2024-11-18
> **Rebuttal**
>
> We thank the reviewer for their comments and address their feedback below.
> > Most of the important details are pushed into appendix. The main body only contains one useful theorem which identifies a certain condition where condition 1) of weak to strong generalization holds. Setting for condition 2) and multi class settings are merely mentioned as claims.
> - Condition 2) is essentially Desiderata 2.ii, which is not a core part of the weak-to-strong generalization phenomenon. We mention this merely as a claim because, it essentially just reduces to checking the conditions from Theorem 3.1, i.e. that $\tau_{\mathsf{strong}} < 0$. We have updated Remark 3.4 to make this more explicit.
> - Regarding the multilabel/multiclass setting (Theorem 3.5), as stated in the text, it requires some additional setup, so due to space constraints we could not elaborate much further on it formally. We will add some comments to convey the intuition of the proof and how it reduces to the binary classification analysis.
>
> > The main body also does not provide proof sketch or provide insights into the proof of the theorem.
> - In the revision, we have moved the proof sketch back into the main text. The brief idea is that, we can analyze the SNR of the true direction $v_*$ when the model is trained on labels generated by another direction $w$. We can use this high level strategy to analyze the generalization of the weak model, as well as the generalization of the weak-to-strong model.
>
> > Reduce the introduction - it currently spans 2 pages.
> - We have condensed the introduction, while still attempting to be fair and thorough with our literature review.
>
> > Figure 1 is useless
> - We have removed Figure 1.
>
> > The section on data model was not particularly needed. Page 5 and 6 can be compressed into 1 or 2 paragraphs.
> - We believe that since the weak-to-strong generalization setup is slightly nonstandard, it can be a little confusing if readers are unfamiliar with the concept. Also, since the result crucially relies on the data assumptions, it was worth being more thorough with the assumptions (perhaps at the risk of being slightly verbose). We have attempted to slightly trim down on the more redundant parts, but think that compressing much more would hamper the paper’s readability.
>
> > Include some experiments in main body.
> - We have moved some of the MNI experiments from the appendix back into the main text in Figure 2.

---

> > ### Comment · Reviewer_vRgg · 2024-12-02
> >
> > Thank you for addressing the concerns. I would like to keep the current score given the limited utility of the developed theory.

---

### Official Review · Reviewer_Ns5A · 2024-11-04

**Soundness:** 4
**Presentation:** 3
**Contribution:** 4
**Rating:** 6
**Confidence:** 3

**Summary:**

The paper investigates weak-to-strong generalization in the setting of an overparameterized spiked covariance model with Gaussian covariates. The paper identifies an asymptotic phase transition between successful and unsuccessful generalization.

**Strengths:**

The math appears correct to me; the problem is significant, and desiderata 1 and desiderata 2 make sense.

**Weaknesses:**

The paper is rather technical, and the clarity could be improved significantly to make it more readable. (see questions)

**Questions:**

1. The main setup is quite confusing to me. The paper first states that "$f_{weak} \in \mathbb{R}^d$" is the object we learn. Normally, the model is a function, not a vector, so this was not immediately clear. It is defined later in line 347 how we learn $ f $, which is quite far from where it was introduced (line 184). It would be better to define that we train $f$ by MNI earlier.

2. In line 201, it says, "As a consequence of our main results in Section 3, we will show that the above desiderata are achievable in a simple toy model; see Theorem 3.3 for a formal statement." However, Theorem 3.3 only considers desiderata 1.2 and 2.1, not the entirety of the desiderata.

3. What is "$t$" in Equation (3) of Theorem 3.1?

4. The notation $ u, p, q, r $ used is not very intuitive, and it makes the result difficult to interpret. Is there a simpler way to rephrase the result?

---

> ### Author Response · Authors · 2024-11-18
> **Rebuttal**
>
> We thank the reviewer for their feedback. Below, we address their comments.
> > The main setup is quite confusing to me. The paper first states that "$f_{\mathsf{weak}} \in \mathbb{R}^d$" is the object we learn. Normally, the model is a function, not a vector, so this was not immediately clear. It is defined later in line 347 how we learn $f$, which is quite far from where it was introduced (line 184). It would be better to define that we train $f$ by MNI earlier.
> - We apologize for the confusion with the notation. We have uploaded an updated version where we clarify this, including clarifying how $f$ is trained by MNI earlier. Our goal with the discussion around Line 184 was to encapsulate many different w2s training schemes, but we agree it is helpful to keep a concrete algorithm in mind.
>
> > In line 201, it says, "As a consequence of our main results in Section 3, we will show that the above desiderata are achievable in a simple toy model; see Theorem 3.3 for a formal statement." However, Theorem 3.3 only considers desiderata 1.2 and 2.1, not the entirety of the desiderata.
> - The two equations in Lines 421-422 contain the conditions for the additional Desiderata being satisfied. In the revision, we have stated more explicitly that Desiderata 1.i-1.iii are all satisfied, and changed the tags to make it more intuitive. In addition, Remark 3.4 discusses Desiderata 2.i and 2.ii. We felt it would be distracting to focus on the bonus desiderata too much, so we moved it to the Remark. We  updated the writing near Line 201 to reference Remark 3.4 regarding the bonus desiderata.
>
> > What is $t$ in Equation (3) of Theorem 3.1?
> - $t \in [0, s)$ controls the number of label classes in the multiclass problem: $k = n^t$, following Definition 2. We have updated the wording of the theorem to make this more explicit.
>
> > The notation $u,p,q,r$ used is not very intuitive, and it makes the result difficult to interpret. Is there a simpler way to rephrase the result?
> - We apologize for the confusion. The reason we chose this type of notation is that prior work in this area has used similar conventions (see, e.g., [1-4]). We thought it might be easier to work in log space (where conditions are additive) as opposed to multiplicative, but we can try to add a couple sentences to rephrase things.
>
> [1] Wang, K., & Thrampoulidis, C. (2022). Binary classification of gaussian mixtures: Abundance of support vectors, benign overfitting, and regularization. SIAM Journal on Mathematics of Data Science, 4(1), 260-284.
>
> [2] Wang, K., Muthukumar, V., & Thrampoulidis, C. (2021). Benign overfitting in multiclass classification: All roads lead to interpolation. Advances in Neural Information Processing Systems, 34, 24164-24179.
>
> [3] Muthukumar, V., Narang, A., Subramanian, V., Belkin, M., Hsu, D., & Sahai, A. (2021). Classification vs regression in overparameterized regimes: Does the loss function matter?. Journal of Machine Learning Research, 22(222), 1-69.
>
> [4] Wu, D., & Sahai, A. (2024). Precise asymptotic generalization for multiclass classification with overparameterized linear models. Advances in Neural Information Processing Systems, 36.

---

### Meta-Review · Area_Chair_d7uG · 2024-12-25

**Metareview:**

Motivated by recent work by Burns et al., this paper identifies a stylized setting where one can show weak to strong generalization theoretically. The weak model is trained on 'weak features' and this model is used to generate pseudolabels for more (unlabeled data). The strong model is trained with 'strong features' on pseudolabels. The authors show that in this case: (i) the weak model has not much better than random accuracy, and (ii) the strong model has almost optimal accuracy. Furthermore an additional condition that the strong model can fully represent the required model is satisfied. The paper focuses on multi-class classification with min-l2-norm interpolation. While it is nice that there are theoretical results, the reviewers have questioned whether the model is realistic enough to actually capture the phenomena observed in prior empirical work. There are also lots of assumptions that are not always easy to interpret.

**Additional Comments On Reviewer Discussion:**

The reviewers engaged well with the authors.

---

### Decision · Program_Chairs · 2025-01-22

Accept (Poster)